# AWT: Transferring Vision-Language Models via Augmentation, Weighting, and Transportation

**Yuhan Zhu**[1]    **Yuyang Ji**[1]    **Zhiyu Zhao**[1,2]    **Gangshan Wu**[1]    **Limin Wang**[1,2*]

[1]State Key Laboratory for Novel Software Technology, Nanjing University
[2]Shanghai AI Laboratory
https://github.com/MCG-NJU/AWT

## Abstract

Pre-trained vision-language models (VLMs) have shown impressive results in various visual classification tasks. However, we often fail to fully unleash their potential when adapting them for new concept understanding due to limited information on new classes. To address this limitation, we introduce a novel adaptation framework, AWT (Augment, Weight, then Transport). AWT comprises three key components: augmenting inputs with diverse visual perspectives and enriched class descriptions through image transformations and language models; dynamically weighting inputs based on the prediction entropy; and employing optimal transport to mine semantic correlations in the vision-language space. AWT can be seamlessly integrated into various VLMs, enhancing their zero-shot capabilities without additional training and facilitating few-shot learning through an integrated multimodal adapter module. We verify AWT in multiple challenging scenarios, including zero-shot and few-shot image classification, zero-shot video action recognition, and out-of-distribution generalization. AWT consistently outperforms the state-of-the-art methods in each setting. In addition, our extensive studies further demonstrate AWT's effectiveness and adaptability across different VLMs, architectures, and scales.

## 1   Introduction

Recent advances in vision-language models (VLMs) [1–8], which undergo extensive pre-training on web-scale image-text pairs, have exhibited remarkable success in various classification tasks. VLMs are trained to associate images with relevant textual descriptions. In the standard protocol (Fig. 1(a)), raw images and class names are projected into a joint vision-language embedding space, where the class with the shortest distance to the image representation is selected as the prediction result.

However, directly using raw images and class names in testing has limitations [1, 9]. Visually, the broad scope of pre-training compels VLMs to analyze all image elements, lacking capability of focusing on specific interested regions. For instance, a model might miss critical facial features of a cat while unnecessarily focusing on irrelevant elements like "bench" and "grass" (Fig. 1(a)). Textually, since VLM pre-training associates visual elements with diverse and rich textual descriptions (*e.g.*, colors and textures), merely using class names during test falls short of capturing the full spectrum of visual content. To enhance input effectiveness, the literature focuses on post-training prompts [9–33] (Fig. 1(b)) that provide contextual cues, thereby helping the model in prioritizing relevant features, such as cat's attributes. However, this approach often depends on the availability of training resources, which may not be always practical.

In this study, we are interested in enhancing inputs for better adaptation of VLMs **without** training prompts. We advocate for data augmentation as a simple yet effective strategy, as depicted in Fig. 1(c).

---

*Corresponding author: lmwang@nju.edu.cn

38th Conference on Neural Information Processing Systems (NeurIPS 2024).

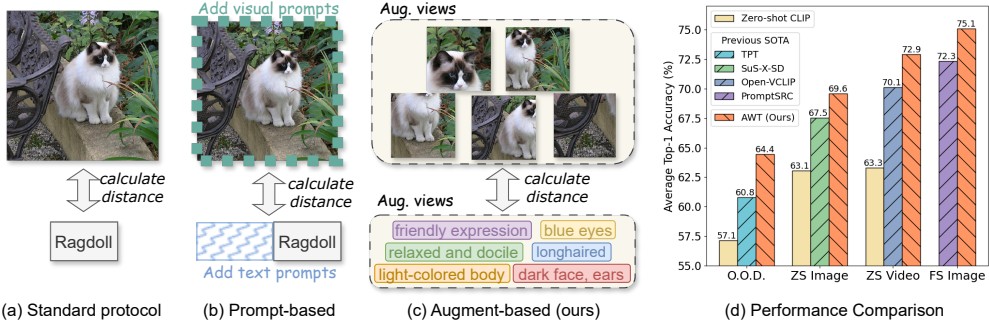

Figure 1: (a) Standard protocol directly calculates distances between raw images and class names in the joint V-L space. (b) Prompt-based methods enhance inputs with post-trained visual or textual prompts to provide the task-specific context. (c) Augment-based method enriches raw inputs with image transformations and class descriptions, requiring no additional training. Upon this, we propose AWT, which considers both intra-modal importance variations and cross-modal semantic correlations. (d) AWT is evaluated against SOTA methods across four tasks: zero-shot and few-shot image classification, out-of-distribution generalization, and zero-shot video action recognition.

Techniques like random resized cropping and image flipping enrich the input with varied and multiscale perspectives, while detailed textual descriptions for each class provide richer visual narratives. Although manually crafting diverse descriptions for each class is expensive, employing Large Language Models (LLMs) [34–36] presents an efficient alternative.

Nonetheless, several challenges remain. First, the intra-modal importance of each augmented image and description needs assessment, as not all views contribute equally to class recognition—some may be irrelevant background elements or non-visual descriptors such as the cat's personality. Second, the inter-modal interaction requires consideration, as descriptions such as "dark face" or "light-colored body" might have direct semantic correlations with some image crops (Fig. 1(c)).

To tackle these challenges, we propose AWT, a novel framework that **augments** raw inputs into diverse views, **weights** view importance in each modality dynamically, and **transports** semantic correlations across modalities. Initially, AWT augments raw inputs via image transformations and LLMs. Subsequently, it weights the importance of each view on the fly based on its prediction entropy, as more confident predictions typically indicate higher accuracy [37]. This method allows AWT to identify and prioritize significant views, and adjust the importance distribution dynamically according to the task-specific context (*e.g.*, candidate class names). AWT then formulates the image-text distance calculation as an optimal transport problem [38, 39], considering each augmented view as a quantity of sand. The importance assessed for each view determines the mass of its corresponding sand pile, and distances are calculated using cosine similarity. This formulation can effectively discover cross-modal correlations by solving the optimal transport problem—which minimizes the effort required to transport sand from one modality to another. Additionally, generating class descriptions from LLMs using a simple prompt like "Describe a {class}." often results in overly generic descriptions. Inspired by chain-of-thought approach [40], we introduce a two-step, dataset-aware prompting method. This approach encourages LLMs to produce class descriptions that are both diverse and dataset-relevant.

We implement AWT using the CLIP model [1] and evaluated its performance across 21 datasets covering four challenging tasks: zero-shot and few-shot image classification, out-of-distribution generalization, and zero-shot video action recognition. As shown in Fig. 1(d), AWT consistently surpasses the existing state-of-the-art methods in each setting. Our extensive analysis further examines AWT's flexibility with diverse architectures, its scalability with different model sizes, and its potential applicability to other VLMs.

## 2 Related Work

**Vision-Language Models.** Leveraging the extensive pre-training on web-scale text-image pairs, vision-language models (VLMs) such as CLIP [1] and ALIGN [6] excel in acquiring versatile

representations that span multiple modalities. These models adeptly embed texts and images into a shared vision-language feature space, enabling the proximity of inputs with analogous semantics. The inherent flexibility of natural language allows VLMs to be effectively utilized across a wide range of open-set tasks including image classification [1, 9, 13], object detection [41–43], image generation [44, 45], video action recognition [46–48]. However, such general-purpose models often fail to focus on task-specific details, which can result in sub-optimal performance. This study aims to overcome this limitation by proposing a novel adaptation framework, namely AWT, for VLMs.

**Adapt VLMs to downstream tasks.** Direct adaptation of pre-trained VLMs to downstream tasks often results in suboptimal and unstable performance [9]. To overcome this, the existing literature has primarily focused on the use of post-training to enrich task context. This includes strategies such as few-shot prompt learning [9, 11, 12, 29, 32], cross-dataset prompt generalization [13–22, 25–27, 30, 31, 33], unsupervised prompt tuning [23, 24, 28], test-time prompt tuning [10, 49–54], and adapter tuning [55–59]. Conversely, other approaches aim to augment inputs using various resources such as the WordNet relationship hierarchy [60], Large Language Models [61–64], or Stable Diffusion models [50, 65, 66]. Nonetheless, these methods mainly enhance only one modality. In contrast, our study innovatively applies augmentation to both visual and textual modalities and addresses significant challenges in dual-modality augmentation scenarios.

**Optimal Transport (OT).** Optimal transport (OT), originating from the Monge problem [38] in the eighteenth century, serves as a metric for quantifying the distance between mathematical entities [67] while considering their intricate geometric structures [39]. Historically rediscovered in various forms, OT first gained fame in computer vision under the name of earth mover's distances [68]. The development of efficient approximate solvers [69] has recently propelled a resurgence in OT's popularity, broadening its utility across multiple domains, including object detection [70, 71], domain adaptation [72–74], generative modeling [75–78], semantic correspondence [79], point clouds [80–82], prompt learning [83–85] and video understanding [86, 87]. Of particular relevance to our study are PLOT [83] and Wang et al. [85], which leverage OT for fine-grained prompt learning to enhance VLMs. Distinct from these two studies, our research diverges by eschewing the need for additional training resources, opting instead for an augmentation-based direction.

## 3 Methodology

### 3.1 Preliminaries

**Contrastive Language-Image Pre-training (CLIP).** CLIP [1] integrates dual encoders—an image encoder $f(\cdot)$ and a text encoder $g(\cdot)$—to map images and textual descriptions into a shared vision-language (V-L) embedding space. CLIP is designed to minimize the cosine distance between embeddings of semantically related image-text pairs. Thanks to the flexibility of natural language, CLIP enables direct application to classification tasks without the need for task-specific training. For instance, given an image $X \in \mathbb{R}^{3 \times H \times W}$ and a set of candidate class names $\{t_i\}_{i=1}^C$, where $C$ denotes the class count. CLIP computes the embeddings $\boldsymbol{I} \in \mathbb{R}^d$ for the image and $\{\boldsymbol{e}_i\}_{i=1}^C \in \mathbb{R}^{C \times d}$ for all class names, where $d$ is the feature dimension. Subsequently, the classification probability for image $X$ being of class $t_i$ can be formulated as:

$$p\left(t_i \mid X\right) = \frac{\exp\left(\cos\left(\boldsymbol{e}_i, \boldsymbol{I}\right)/\tau\right)}{\sum_{j=1}^C \exp\left(\cos\left(\boldsymbol{e}_j, \boldsymbol{I}\right)/\tau\right)}, \tag{1}$$

where $\tau$ is a temperature parameter.

**Optimal Transport (OT).** Optimal transport (OT) theory, originating from the Monge problem [38], provides a framework for structural distance measurements. This theory conceptualizes scenarios such as relocating sand at a construction site with the goal of minimizing effort. Mathematically, the initial and target distributions of sands are modeled as discrete measures:

$$\alpha = \sum_{i=1}^N \mathbf{a}_i \delta_{x_i} \quad \text{and} \quad \beta = \sum_{j=1}^M \mathbf{b}_j \delta_{y_j}, \tag{2}$$

where $\delta_{x_i}$ denotes the Dirac with a concentrated mass $\mathbf{a}_i$ centered at $x_i$, and similarly for $\beta$. Here, $N$ and $M$ represent the number of source and target locations, respectively. The cost of transporting

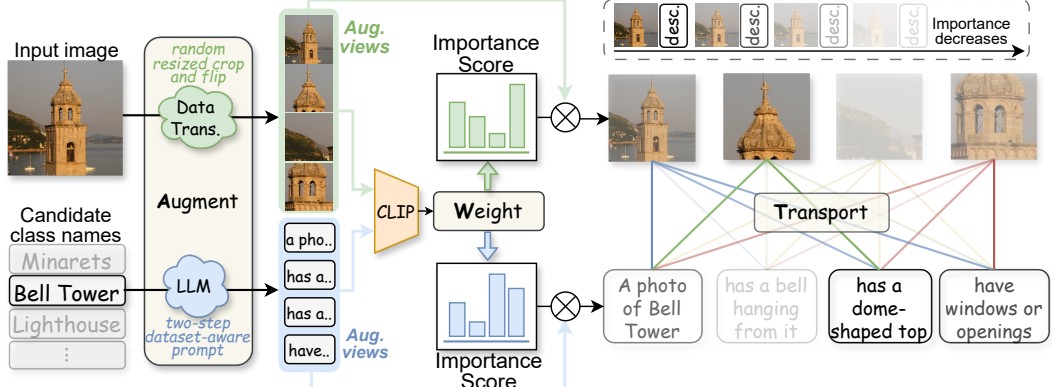

Figure 2: **Pipeline of AWT: Augment, Weight, then Transport.** Given an image and candidate class names, we first augment each input into diverse views. These views are then fed into the CLIP model to obtain coarse predictions. To assess the importance of each view, we use prediction confidence as a proxy and introduce an entropy-based weighting mechanism. Next, we measure the distance between image-text view sets by solving an optimal transport (OT) problem. Finally, the resulting OT distance is used to represent the distance between the input image and each class name.

sands from any source location $x_i$ to any target location $y_j$ is given by the cost function $c(x_i, y_j)$. To extend the application to broader and more intricate scenarios, *e.g.*, cross-modal correlation, the Kantorovich relaxation [88] is employed. This relaxation introduces flexibility in the transport plan and ensures symmetric transport solutions. The transport plan $\mathbf{P} \in \mathbb{R}_+^{N \times M}$, where element $\mathbf{P}_{i,j}$ indicates the mass transported from $x_i$ to $y_j$, must satisfy the constraints:

$$\mathbf{U}(\mathbf{a}, \mathbf{b}) \stackrel{\text{def.}}{=} \left\{ \mathbf{P} \in \mathbb{R}_+^{N \times M} : \mathbf{P}\mathbb{1}_M = \mathbf{a} \quad \text{and} \quad \mathbf{P}^{\mathrm{T}}\mathbb{1}_N = \mathbf{b} \right\}. \tag{3}$$

Kantorovich's formulation seeks to minimize the total transportation cost:

$$\mathcal{L}_c(\alpha, \beta) \stackrel{\text{def.}}{=} \min_{\mathbf{P} \in \mathbf{U}(\mathbf{a}, \mathbf{b})} \langle \mathbf{C}, \mathbf{P} \rangle \stackrel{\text{def.}}{=} \sum_{i,j} \mathbf{C}_{i,j} \mathbf{P}_{i,j}, \tag{4}$$

where $\mathbf{C}_{i,j} = c(x_i, y_j)$ defines the cost matrix.

## 3.2 AWT: Augment, Weight, then Transport

Pre-trained VLMs often underperform when adapted to new concepts due to insufficient information about new classes. Moreover, their extensive pre-training scope leads them to analyze all elements of an image, causing them to miss contextually important cues crucial for specific downstream applications. To overcome these limitations, we introduce a novel framework, termed AWT (Augment, Weight, then Transport), to enhance the adaptability of VLMs without additional training. The AWT framework, as depicted in Fig. 2, consists of three critical components: augmenting raw inputs to generate diverse and content-rich views, weighting the significance of these views within each modality, and transporting semantically correlated elements across modalities.

### 3.2.1 Augment Raw Inputs

The augmentation process begins with an image $X \in \mathbb{R}^{3 \times H \times W}$ and the class name set $\{t_i\}_{i=1}^C$, aiming to transform these inputs into various views that offer different perspectives and details.

For visual augmentation, we apply standard data augmentation including random resized cropping and random flipping to produce a set of varied views $\{X^n\}_{n=1}^{N+1}$. This set includes $N$ augmented images alongside the original (denoted as the $0$ index), enriching the input with diverse and multiscale perspectives. An illustrative example is shown in Fig. 2.

To enrich the textual modality, we utilize Large Language Models (LLMs) to generate class descriptions. Typical prompts like "`Describe a {class}.`" often result in descriptions that are either vague—lacking in specific visual details—or contextually misaligned. For instance, in contexts

such as classifying sketches, generic descriptions of a category may not correspond well with the sketch images. To address this, we adopt a two-step, dataset-aware prompt strategy. Initially, we prompt LLMs to generate multiple questions that probe different aspects of the category, which is crucial for eliciting detailed and varied descriptions. To ensure the queries are aligned with the visual content, we incorporate a dataset-level description into the initial prompts. Specifically, we start by asking LLMs to "`Generate questions to classify images from a dataset, which {dataset descirption}.`". Using the dataset-related questions generated from the first step, we proceed to the second step where these questions are combined with the specific class name to obtain tailored descriptions. The set of augmented views for each class $t_i$ is denoted as $\{t_i^m\}_{m=1}^{M+1}$, including an additional view formed by the raw class name. This method ensures both diversity in the descriptions and their relevance to the visual content. More details can be found in Appendix C.2.

### 3.2.2  Weight Augmented Views

Following augmentation, it is essential to assess the significance of each augmented view, as not all views contribute equally to classification. Some views may be critical while others might be less informative or even noisy. To address this variation, we introduce an entropy-based weighting mechanism to quantify each view's importance. Our key insight is that the impact of a view on classification confidence—a metric often correlated with accuracy [37]—can serve as a proxy for its importance. A view that significantly enhances classification confidence is considered more vital.

To assess the importance of $n$-th image view $X^n$, we maintain a constant text context and compute the averaged embedding for each class as $\{\bar{e}_i = \frac{1}{M+1}\sum_{m=1}^{M+1} e_i^m\}_{i=1}^C$, where $e_i^m$ is the CLIP embedding of $t_i^m$. The classification probability $p(t \mid X^n)$ is then calculated using the image embedding $\boldsymbol{I}^n$ and text embeddings $\{\bar{e}_i\}_{i=1}^C$, following Eq. (1). Predictive confidence is then quantified using the entropy formula $H_n(t) = -\sum_t p(t \mid X^n) \log p(t \mid X^n)$. Lower entropy indicates higher confidence, allowing us to evaluate view importance through the negative entropy as follows:

$$\mathbf{a}_n = \frac{\exp\left(-H_n(t)/\gamma_v\right)}{\sum_{j=1}^{N+1} \exp\left(-H_j(t)/\gamma_v\right)}, \quad n = 1, \ldots, N+1, \tag{5}$$

where $\gamma_v$ is a temperature parameter adjusting the distribution's sharpness.

Similarly, to determine the importance of $m$-th description for $i$-th class, *i.e.*, $t_i^m$, we calculate the classification probability $p_m^i(t \mid X^0)$, with the image embedding $\boldsymbol{I}^0$ and text embeddings $\{e_i^m\} \cup \{\bar{e}_j\}_{j=1,j\neq i}^C$. The classification entropy is given by $H_m^i(t) = -\sum_t p_m^i\left(t \mid X^0\right)\log p_m^i\left(t \mid X^0\right)$. We then calculate the importance scores for all descriptions within the $i$-th class as follows:

$$\mathbf{b}_m^i = \frac{\exp\left(-H_m^i(t)/\gamma_t\right)}{\sum_{k=1}^{M+1} \exp\left(-H_k^i(t)/\gamma_t\right)}, \quad m = 1, \ldots, M+1, \tag{6}$$

where $\gamma_t$ is the temperature parameter. This entropy-based weighting mechanism ensures the prioritization of the contextually significant views. By dynamically adjusting the importance based on the direct impact on classification confidence, the augmented view sets can be well-prepared for the optimal transport process.

### 3.2.3  Transport Across Modalities

Our primary goal is to precisely measure the distance between an image and its candidate names. Through the augmentation, we have transformed each original image or class name into a set of augmented views. Typically, the distance between these sets is measured by simply averaging the embeddings within each set. However, such practice often fails to capture the dynamic correlation across modalities. Consider the scenario depicted in Fig. 2, where specific textual descriptions such as "has a dome-shaped top" might correlate directly with certain image crops. The conventional averaging strategy typically overlooks these intuitive and meaningful correlations.

To address this issue, we propose a novel approach by formulating distance measurement as an optimal transport (OT) problem, which facilitates richer interactions between modalities. We model each view within the V-L space as a mass located at its embedding position:

$$\alpha = \sum_{n=1}^{N+1} \mathbf{a}_n \delta_{\boldsymbol{I}^n} \quad \text{and} \quad \left\{\beta^i = \sum_{m=1}^{M+1} \mathbf{b}_m^i \delta_{e_i^m}\right\}_{i=1}^C. \tag{7}$$

Table 1: **Zero-shot image classification.** We report top-1 accuracy (%) for each dataset. The "Train" column indicates whether the methods necessitate additional training (including test-time training). Numbers in grey indicate that the method was trained on ImageNet and is therefore not zero-shot.

| | Train | IN-1k[89] | Flowers[90] | DTD[91] | Pets[92] | Cars[93] | UCF[94] | Cal101[95] | Food[96] | SUN[97] | Aircraft[98] | SAT[99] | Birds[100] | Cal256[101] | CUB[102] | Avg(14) |
|---|---|---|---|---|---|---|---|---|---|---|---|---|---|---|---|---|
| CLIP [1] | ✗ | 66.74 | 67.44 | 44.27 | 88.25 | 65.48 | 65.13 | 93.35 | 83.65 | 62.59 | 23.67 | 42.01 | 42.80 | 82.50 | 54.90 | 63.06 |
| CoOp [9] | ✓ | 71.51 | 68.71 | 41.92 | 89.14 | 64.51 | 66.55 | 93.70 | 85.30 | 64.15 | 18.47 | 46.39 | 41.43 | 82.91 | 53.18 | 63.42 |
| CoCoOp [13] | ✓ | 71.02 | 71.88 | 45.73 | 90.14 | 65.32 | 68.21 | 94.43 | 86.06 | 67.36 | 22.94 | 45.37 | 43.75 | 85.39 | 56.09 | 65.26 |
| MaPLe [15] | ✓ | 70.72 | 72.23 | 46.49 | 90.49 | 65.57 | 68.69 | 93.53 | 86.20 | 67.01 | 24.74 | 48.06 | 44.06 | 85.58 | 57.18 | 65.75 |
| PLOT++ [83] | ✓ | 72.48 | 69.10 | 38.42 | 90.49 | 61.20 | 68.94 | 91.32 | 86.07 | 61.59 | 24.84 | 49.90 | 36.37 | 84.30 | 48.58 | 63.11 |
| POMP [22] | ✓ | 70.16 | 72.72 | 44.44 | 89.05 | 66.70 | 68.44 | 94.65 | 86.28 | 67.27 | 25.47 | 52.65 | 43.94 | 86.76 | 56.92 | 66.10 |
| ProVP-Ref [32] | ✓ | 71.14 | 71.62 | 45.97 | 91.58 | 65.29 | 67.72 | 93.79 | 86.17 | 66.29 | 24.51 | 51.95 | – | – | – | 66.91 |
| TPT [49] | ✓ | 68.98 | 68.98 | 47.75 | 87.79 | 66.87 | 68.04 | 94.16 | 84.67 | 65.50 | 24.78 | 42.44 | 44.56 | 85.71 | 56.97 | 64.80 |
| DiffTPT [50] | ✓ | 70.30 | 70.10 | 47.00 | 88.22 | 67.01 | 68.22 | 92.49 | **87.23** | 65.74 | 25.60 | 43.13 | – | – | – | 65.91 |
| PromptAlign [52] | ✓ | 71.44 | 72.39 | 47.24 | 90.76 | 68.50 | 69.47 | 94.01 | 86.65 | 67.54 | 24.80 | 47.86 | 45.28 | 86.05 | 57.90 | 66.42 |
| Self-TPT-v [10] | ✓ | 72.96 | 71.79 | 49.35 | 91.26 | 68.81 | 69.50 | 94.71 | 85.41 | 68.18 | 27.57 | 51.91 | – | – | – | 68.31 |
| CuPL [63] | ✗ | 69.62 | 71.30 | 44.56 | 89.13 | 65.29 | 66.83 | 92.98 | 86.11 | 62.59 | 24.90 | 47.84 | 41.24 | 86.30 | 56.28 | 64.64 |
| VisDesc [61] | ✗ | 68.55 | 70.85 | 44.98 | 88.85 | 64.08 | 67.12 | 94.60 | 85.05 | 67.99 | 24.30 | 54.84 | 43.64 | 87.16 | 56.59 | 65.61 |
| WaffleCLIP [62] | ✗ | 68.81 | 72.35 | 45.21 | 89.95 | 63.57 | 67.19 | 94.02 | 86.68 | 67.23 | 25.39 | 55.07 | 43.92 | 87.04 | 57.17 | 65.97 |
| SuS-X-SD [65] | ✗ | 69.88 | 73.81 | 54.55 | 90.57 | 66.13 | 66.59 | 93.96 | 86.08 | 67.73 | 28.68 | 57.49 | 45.53 | 87.45 | 57.11 | 67.54 |
| AWT | ✗ | 71.32 | 75.07 | 55.56 | 92.53 | 69.93 | 72.51 | 95.54 | 85.54 | 70.58 | 29.22 | 58.61 | 48.75 | 88.84 | 60.20 | 69.59 |

Here, the importance weight of each view, derived from Eqs. (5) and (6), determines the mass allocation. The transportation cost between any two points (*e.g.*, an image and a textual description) is quantified using the cosine distance between their embeddings, $\mathbf{C} = 1 - \cos(\boldsymbol{I}, \boldsymbol{e})$, which serves as an intuitive measure of semantic similarity. The goal of optimal transport is to minimize the total cost of transporting mass from visual modality into textual modality. Specifically, the distance between the image view set $\{X^n\}_{n=1}^{N+1}$ and $i$-th class description set $\{t_i^m\}_{m=1}^{M+1}$ is redefined as an OT problem between $\alpha$ and $\beta_i$, as formulated in Eq. (4). We employ Sinkhorn's Algorithm [69] to efficiently approximate the solution, denoted as $\tilde{\mathbf{P}}$. Consequently, the classification probability can be expressed as:

$$p_{\mathrm{OT}}\left(t_i \mid X\right) = \frac{\exp\left(\mathbf{s}_i / \tau\right)}{\sum_{j=1}^{C} \exp\left(\mathbf{s}_j / \tau\right)}, \tag{8}$$

where $\mathbf{s} = \sum_i \sum_j \tilde{\mathbf{P}}_{ij} \left(1 - \mathbf{C}\right)_{ij}$. By employing the optimal transport framework, we ensure that semantically related views receive more attention, enhancing the accuracy and relevance of the classification process.

## 4 Experiments

### 4.1 Zero-shot Image Tasks

**Datasets.** For zero-shot image tasks, we consider image classification and out-of-distribution (OOD) generalization. Our study encompasses 18 datasets that span a wide array of recognition tasks: ImageNet [89], Caltech101 [95] and Caltech256 [101] for generic object recognition, Oxford-Pets [92], StanfordCars [93], OxfordFlowers [90], Food101 [96], FGVCAircraft [98], Birdsnap [100] and CUB [102] for fine-grained classification, SUN397 [97] for scene recognition, DTD [91] for texture classification, EuroSAT [99] for satellite recognition, and UCF101 [94] for action recognition. Besides, four ImageNet variant datasets are involved to assess the model's capability for OOD generalization: ImageNet-A [103], ImageNetV2 [104], ImageNet-R [105] and ImageNet-Sketch [106].

**Competitive methods.** We mainly compare three distinct categories of approaches: 1) Prompt learning methods: These involve additional data to post-train visual or textual prompts, including CoOp [9], CoCoOp [13], MaPLe [15], PLOT++ [83], POMP [22]. 2) Test-time prompt tuning methods: These optimize prompts during inference, such as TPT [49], DiffTPT [50], and PromptAlign [52]. 3) Augment-based method: These use LLMs or diffusion models to augment inputs, including CuPL [63], VisDesc [61], WaffleCLIP [62], and SuS-X-SD [65].

**Implementation details.** We implemented the AWT framework using the CLIP-B/16 model [1]. Image augmentations include random resized cropping and flipping, and class descriptions are generated via GPT-3.5 [35]. We set the number of augmented images $N$ and descriptions $M$ to 50 each. Dataset-level descriptions are provided in Appendix C. For both visual and textual modalities, we configured the importance distribution temperatures at $\gamma_v = 1/2$ and $\gamma_t = 1/2$. The optimal transport problem is approximated using Sinkhorn's Algorithm with an $\epsilon$ of 0.1 [69]. All experiments are conducted on one NVIDIA A100-SXM4-80GB GPU.

**Results.** In Tab. 1, we compare AWT with three categories of CLIP adaptation methods: prompt learning, test-time prompt tuning, and existing augmentation-based methods. Remarkably, without additional training, AWT outperforms all existing methods by a significant margin, achieving state-of-the-art performance on 13 out of 14 datasets and surpassing the previous best results

Table 2: **Out of distribution generalization.**

| Method | IN-A [103] | IN-V2 [104] | IN-R [105] | IN-K [106] | *OOD* |
|---|---|---|---|---|---|
| CLIP [1] | 47.74 | 60.75 | 73.98 | 46.13 | 57.15 |
| TPT [49] | 54.77 | 63.45 | 77.06 | 47.94 | 60.81 |
| DiffTPT [50] | 55.68 | 65.10 | 75.00 | 46.80 | 60.65 |
| CuPL [63] | 50.72 | 63.27 | 77.05 | 49.02 | 60.02 |
| VisDesc [61] | 49.07 | 61.80 | 75.13 | 47.97 | 58.49 |
| WaffleCLIP [62] | 50.78 | 62.54 | 77.49 | 49.10 | 59.98 |
| **AWT** | **60.33** | **65.15** | **80.64** | **51.60** | **64.43** |

by an average accuracy of 2.05%. Further, the out-of-distribution (OOD) generalization capabilities of AWT are detailed in Table 2. Leveraging dataset-aware prompting and a dynamic weighting approach that adjusts in real-time during testing, AWT effectively manages complex scenarios encountered in OOD. Consequently, AWT stands out by delivering the highest performance across all four OOD datasets, surpassing the previous arts by an average accuracy improvement of 3.62%.

### 4.2 Zero-shot Video Tasks

**Setup.** Here, we focus on the zero-shot video action recognition task, using three representative datasets: UCF101 [94], HMDB51 [107], and Kinetics-600 [108]. For UCF101 and HMDB51, we adopt two evaluation protocols: **1) EP1:** test model on all 101 UCF classes and 51 HMDB classes [47, 117], and report the top-1 accuracy. **2) EP2:** evaluate the model using three official splits and averaging the results of each split. The average top-1 accuracy and standard deviation are reported. For Kinetics-600, the top-1 accuracy and

Table 3: **Zero-shot video action recognition.**

| Method | UCF101 [94] | | HMDB51 [107] | | K600 [108] |
|---|---|---|---|---|---|
| | EP1 | EP2 | EP1 | EP2 | |
| ActionCLIP [46] | 77.4 | 77.5±0.8 | 48.0 | 48.2±1.5 | 62.5±1.2 |
| X-CLIP [109] | - | 72.0±2.3 | - | 44.6±5.2 | 65.2±0.4 |
| Ju et al. [110] | - | 69.3±4.2 | - | 44.3±2.2 | 55.8±0.7 |
| Text4Vis [111] | 79.6 | - | 49.8 | - | 68.9±1.0 |
| AIM [112] | 79.0 | 79.4±1.0 | 49.5 | 50.3±0.8 | 66.7±0.5 |
| ST-Adapter [113] | 77.9 | 77.6±0.7 | 50.3 | 51.1±0.6 | 60.2±1.8 |
| Vita-CLIP [114] | - | 75.0±0.6 | - | 48.6±0.6 | 67.4±0.5 |
| ViFi-CLIP [115] | - | 76.8±0.7 | - | 51.3±0.6 | 71.2±1.0 |
| AdaptFormer [116] | 80.5 | 80.3±1.0 | 50.5 | 51.0±0.8 | 67.0±0.4 |
| Open-VCLIP [47] | 83.5 | 83.4±1.2 | 53.2 | 53.9±1.2 | 73.0±0.8 |
| FROSTER [48] | 85.0 | 84.8±1.1 | 54.5 | 54.8±1.3 | 74.8±0.9 |
| **AWT** | **85.4** | **85.2±0.7** | **56.1** | **57.2±0.9** | **76.1±0.6** |

standard deviation are reported on three validation sets split by Chen and Huang [118].

**Implementation details.** To model temporal dynamics, we follow Open-VCLIP [47] to use neighbor-frame attention and fine-tune CLIP on Kinetics-400 [119]. Note that three test subsets of Kinetics-600 have a disjoint class set compared to Kinetics-400. All AWT configurations are the same for zero-shot image tasks except for the visual augmentation, in which we directly use the different sampled temporal and cropped video frames.

**Results.** In Tab. 3, we present a comparison of our AWT with existing CLIP-based zero-shot video action recognition methods. Although AWT was not originally tailored for video tasks, it sets new records in this domain, outperforming the recent state-of-the-art method, FROSTER, by 1.6% and 2.4% on HMDB51, and by 1.3% on Kinetics-600. These results suggest that our AWT framework could be effectively extended to video understanding tasks.

### 4.3 Few-shot Image Tasks

**Setup.** We assessed the few-shot transfer capabilities of our method across 11 datasets: ImageNet [89], Caltech101 [95], OxfordPets [92], StanfordCars [93], OxfordFlowers [90], Food101 [96], FGVCAircraft [98], SUN397 [97], DTD [91], EuroSAT [99], and UCF101 [94]. We trained our model using 1, 2, 4, 8, and 16 shots. Results are averaged over three runs.

**Implementation details.** All AWT configurations are the same as zero-shot image tasks. In this task, we introduced a multi-modal Adapter for efficient learning, inserted after each Multi-Head

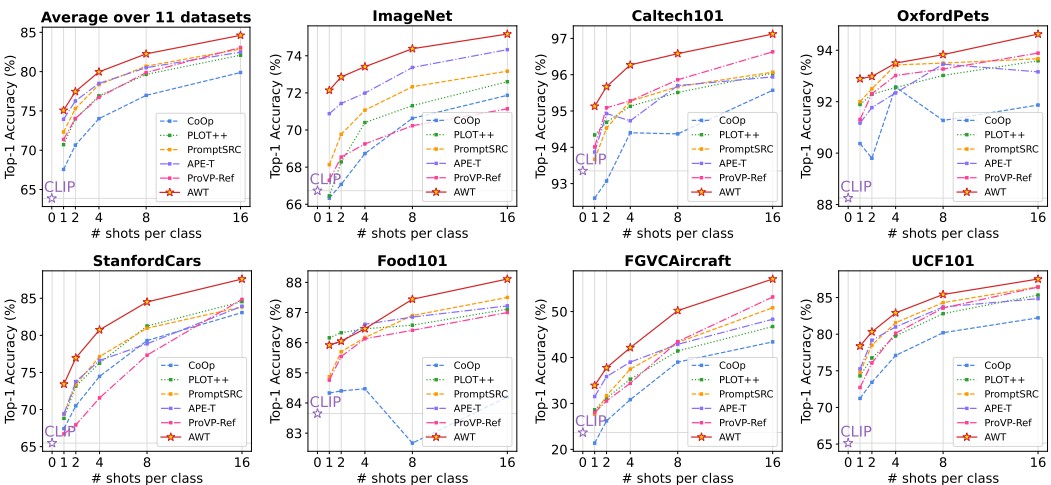

Figure 3: **Few-shot image classification.** We present the average accuracy across 11 datasets and specific accuracy for three datasets. Numerical values can be found at Tab. 11.

Table 4: **Ablation experiments** across 18 image datasets. The default configuration is colored grey.

(a) **Main component analysis**: Augment(A), Weight(W) and Transport(T).

| Step | OOD | Avg.(14) |
|---|---|---|
| Raw inputs | 57.15 | 63.06 |
| A(Img.) | 56.77 | 63.40 |
| A(Name) | 59.76 | 67.36 |
| A(Img.+Name) | 59.92 | 67.13 |
| A+T | 60.48 | 67.80 |
| A+W+T | **64.43** | **69.59** |

(b) Number of **augmented image views** $N$.

| $N$ | OOD | Avg.(14) |
|---|---|---|
| 2 | 61.40 | 68.57 |
| 5 | 62.59 | 68.98 |
| 10 | 63.31 | 69.16 |
| 25 | 64.13 | 69.33 |
| 50 | 64.43 | **69.59** |
| 100 | **64.54** | 69.52 |

(c) Number of generated **class descriptions** $M$.

| $M$ | OOD | Avg.(14) |
|---|---|---|
| 2 | 61.58 | 66.67 |
| 5 | 62.87 | 67.55 |
| 10 | 63.42 | 68.53 |
| 25 | 64.07 | 68.98 |
| 50 | 64.43 | 69.59 |
| 100 | **64.46** | **69.73** |

(d) Different **prompt** methods for LLMs.

| Method | OOD | Avg.(14) |
|---|---|---|
| VisDesc [61] | 63.55 | 67.78 |
| CuPL (base) [63] | 63.57 | 68.87 |
| CuPL (full) [63] | 63.88 | 69.33 |
| Ours | **64.43** | **69.59** |

(e) Weighting **temperature** $\gamma_v$.

| $\gamma_v$ | OOD | Avg.(14) |
|---|---|---|
| 100 | 61.07 | 68.71 |
| 1 | 63.54 | 69.43 |
| 1/2 | **64.43** | **69.59** |
| 1/3 | 64.42 | 69.37 |
| 1/4 | 64.38 | 69.19 |

(f) Weighting **temperature** $\gamma_t$.

| $\gamma_t$ | OOD | Avg.(14) |
|---|---|---|
| 100 | 63.97 | 68.39 |
| 1 | **64.47** | 69.27 |
| 1/2 | 64.43 | **69.59** |
| 1/3 | 64.08 | 69.50 |
| 1/4 | 63.80 | 69.38 |

Self-Attention and MLP block in every layer. We adopt a distillation technique similar to [21] to prevent overfitting. Further details about the Adapter are available in Appendix C.

**Results.** In Fig. 3, we compare the performance of AWT with existing methods in the few-shot transfer learning task. Impressively, AWT surpasses the previous state-of-the-art by average accuracies of 2.76%, 2.16%, 1.62%, 1.57%, and 1.75% for 1, 2, 4, 8, and 16 shots respectively. Notably, on the ImageNet dataset, AWT significantly outperforms all prior methods. While PLOT++ also utilizes optimal transport, it is limited to local image features and class names, neglecting the multiscale image perspectives and the rich textual semantics, leading to suboptimal transfer capabilities. In contrast, AWT leverages diverse augmented views, effectively maintaining intra-modal importance trade-offs while establishing dynamical cross-modal correlations, achieving superior few-shot performance.

### 4.4 Ablation Study

**Main component analysis.** In Tab. 4a, we analyze the key components of AWT. Initially, we augment raw inputs and apply basic ensembling. The results (rows two and four) show that directly augmenting images is ineffective, likely due to background image crops. Conversely, textual enhancements significantly boost performance, thanks to our carefully designed prompt strategies for LLMs. We then shift from basic ensembling to optimal transport (OT), leading to consistent improvements across two tasks. However, the full potential of OT is not realized due to ineffective mass (*i.e.*,

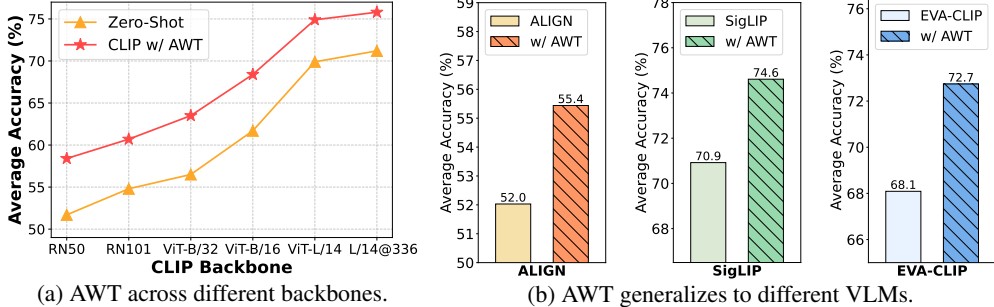

(a) AWT across different backbones.
(b) AWT generalizes to different VLMs.

Figure 4: **Versatility analysis of AWT.** Average top-1 accuracy (%) on 18 image datasets is reported.

importance) weighting. By incorporating our entropy-based weighting method, which accurately assesses the importance of each view, we again achieve substantial performance gains.

**Number of augmentation views.** We present the study on the augmented view quantities for both visual and textual sides in Tabs. 4b and 4c, respectively. The results clearly demonstrate that performance tends to increase with the number of views. Our findings suggest that about 50 views per modality are sufficient to achieve decent performance. The number of augmentation views is crucial for AWT's effectiveness. Given that AWT is augmentation-driven, this correlation is intuitive. However, increasing the augmented view quantities can also lead to higher computational costs during inference, we also include an efficiency-performance trade-off analysis in Fig. 9.

**LLM prompting strategy.** We evaluated the effectiveness of our LLM prompting strategy, detailed in Tab. 4d. Our method is compared with two established approaches: VisDesc [61] and CuPL [63]. VisDesc uses a uniform prompt template across different datasets, while CuPL employs a tailored, dataset-specific manual prompting strategy, enriching context for LLMs. We developed a refined two-step process that enhances context comprehension through dataset-level descriptions and increases diversity by utilizing chain-of-thought queries. Our strategy consistently outperforms the existing methods in both evaluated tasks.

**Temperature in weighting.** We evaluated our entropy-based weighting method by conducting an ablation study on the `softmax` function's temperature parameter (see Eqs. (5) and (6)). A higher temperature creates a more uniform importance distribution. The findings for both modalities are presented in Tabs. 4e and 4f, respectively. Our results reveal that a very high temperature (*e.g.*, 100) leads to suboptimal performance, likely due to insufficient emphasis on contextually significant views. Conversely, lowering the temperature enhances focus on these important views, improving performance. Empirically, a temperature of $1/2$ has been identified as optimal for both modalities. For a clearer understanding of our weighting strategy, visualizations are provided in Appendix A.

### 4.5 Versatility Study

Our AWT is applicable to any VLM using dual encoders to map images and text into a joint space with appropriate distance metrics (*e.g.*. cosine similarity). Therefore, it is crucial to assess AWT's effectiveness across various scenarios. We conduct evaluations with both ResNet [120] and ViT [121] architectures, explore AWT's scalability from ViT-B/32 to ViT-L/14@336, and assess its generalizability across three VLMs: ALIGN [6], SigLIP [122], and EVA-CLIP [123]. We conduct experiments across 18 image datasets and present the results in Fig. 4. Our findings reveal that AWT consistently achieves performance gains in all tested scenarios, highlighting its broad applicability.

### 4.6 Failure Case Analysis

Although AWT has shown success across various datasets and tasks, we identified certain limitations when applying it to the CIFAR datasets. As detailed in Tab. 5, AWT resulted in performance declines of 0.91% and 0.17% on two CIFAR datasets, respectively. To investigate this issue, we analyzed the images produced by the transformations used in our study. We discovered that with low-resolution images, such as $32 \times 32$ pixels, the random resized crop operation tends to overly blur the images,

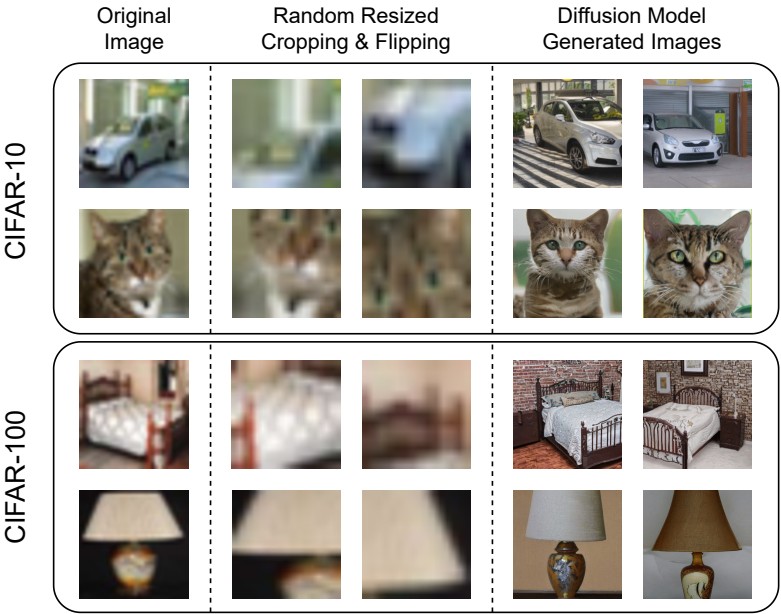

Figure 5: **Comparison of image augmentation techniques** on low-resolution images. We present images from the CIFAR-10/100 datasets, where each image is $32 \times 32$ pixels. The comparison includes images generated by traditional image transformations and DALL·E 2.

Table 5: **Failure case analysis.** We focus on low-resolution datasets CIFAR-10 and CIFAR-100 and evaluate the effectiveness of AWT when equipped with different image augmentation techniques.

| Method | Image augmentation | CIFAR10 [124] | CIFAR100 [124] |
|---|---|---|---|
| CLIP [1] | - | 90.16 *(baseline)* | 67.78 *(baseline)* |
| AWT | Random resized crop and flip | 89.25 (0.91) ↓ | 67.61 (0.17) ↓ |
| AWT | DALL·E 2 [44] | 92.30 (2.14) ↑ | 68.69 (0.91) ↑ |

obscuring the objects within them, as illustrated in Fig. 5. To address this, we integrated a diffusion model, specifically DALL·E 2 [44], as a substitute for traditional data augmentations. Fig. 5 shows examples of enhanced image views generated by DALL·E 2, which produce sharper images and offer a variety of perspectives. By incorporating this advanced technique into the AWT framework, we have significantly improved its performance. The updated benchmark results, presented in Tab. 5, demonstrate that AWT now consistently achieves performance gains over baselines.

## 5 Conclusion

In this paper, we have introduced the AWT (Augment, Weight, then Transport) framework, designed to enhance the transferability of pre-trained vision-language models (VLMs). Rather than using raw images and class names directly, our approach enriches the inputs by augmenting them with diverse visual perspectives and detailed class descriptions. We further develop an entropy-based weighting strategy to dynamically prioritize these augmented views and employ optimal transport to measure the cross-modal distance in the structured visual-language space. The AWT framework not only boosts the zero-shot performance of VLMs without the need for additional training but also facilitates few-shot transfer learning via an integrated multimodal adapter module. Our evaluations across four challenging tasks demonstrate that AWT significantly outperforms existing state-of-the-art methods.

**Acknowledgments.** This work is supported by the National Key R&D Program of China (No. 2022ZD0160900), the National Natural Science Foundation of China (No. 62076119), the Fundamental Research Funds for the Central Universities (No. 020214380119), the Nanjing University-China Mobile Communications Group Co., Ltd. Joint Institute, and the Collaborative Innovation Center of Novel Software Technology and Industrialization.

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

# Appendix

The appendix provides supplementary material including additional visualization results, comparative studies, ablation experiments, detailed methodologies, examples of failure cases, and a discussion on the limitations of our research. The contents are structured as follows:

- Visualizations of our weighting and transportation processes (Appendix A).
- Extended experiments and ablation studies (Appendix B).
- Method details and experimental procedures (Appendix C).
- Discussion on societal impacts (Appendix D).
- Discussion on limitations and future research directions (Appendix E).

# A   Visualization

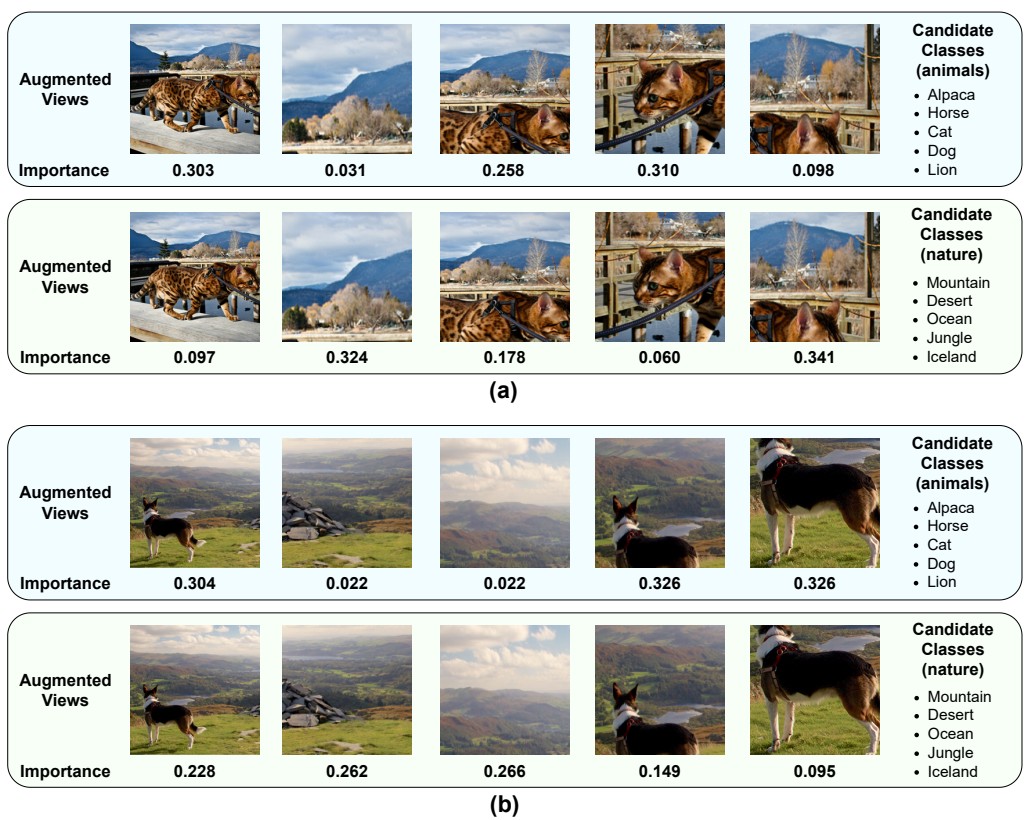

Figure 6: **Visualization of weighting image views.** We show the weights assigned to the same image view set under varying candidate class names. Our dynamic weighting strategy effectively allocates importance to contextually relevant image views.

## A.1   Entropy-based Weight Strategy

Pre-trained VLMs often struggle to adequately focus on contextually significant features due to their open-set design, which requires processing every element within an image. To overcome this limitation, we introduce an entropy-based weighting strategy that dynamically evaluates the importance of each image view based on the prediction entropy. This method allows us to prioritize relevant views while diminishing the impact of less pertinent ones.

**Visualization of weighting image views.**   We have included visualizations to demonstrate the effectiveness of our entropy-based weighting strategy in managing image views, as shown in Fig. 6.

Through these visualizations, we observe two primary benefits: **1) Priority to Significant Views:** The strategy effectively prioritizes image views that are intuitively significant, while efficiently excluding irrelevant background crops. **2) Dynamic weight assignment:** In different contexts, such as varying candidate class names, the strategy dynamically assigns weights, emphasizing views that are contextually important.

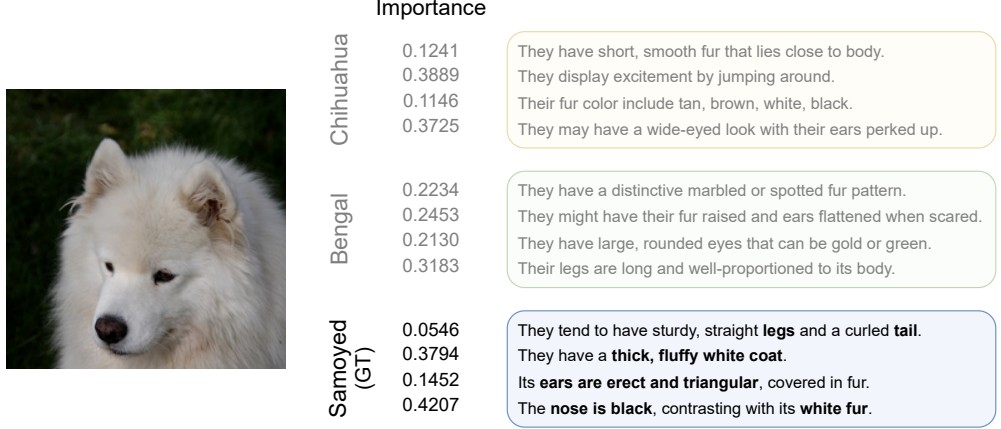

Figure 7: **Visualization of weighting class descriptions.**

**Visualization of weighting class descriptions.** The strategy not only enhances image view management but also optimizes the handling of textual descriptions. In Fig. 7, we observe the following: 1) For descriptions related to the ground-truth class, our strategy prioritizes key textual elements while effectively filtering out irrelevant content. 2) For non-matching classes, the importance levels are relatively uniform, indicating a general irrelevance to the specific image content.

## A.2 Optimal Transport

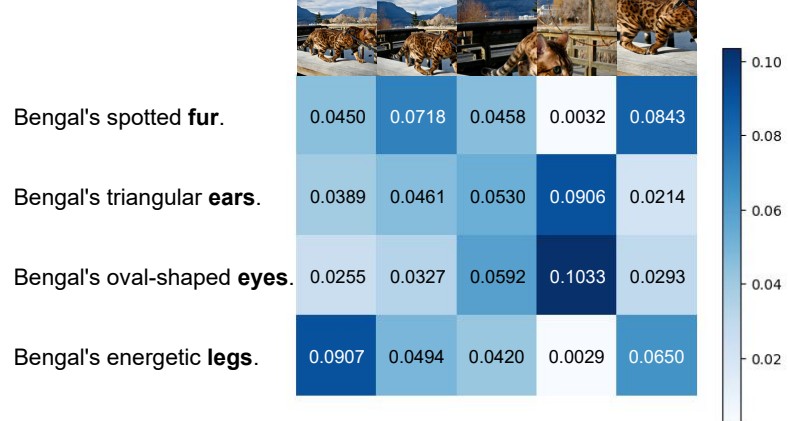

Figure 8: **Visualization of cross-modal correlations** captured by optimal transport.

After augmenting the input image and class names to create diverse views, there is a potential for these views to exhibit direct and meaningful correlations across modalities. For instance, image crops that focus on the eyes of a cat could closely correlate with textual descriptions concerning the eyes. In our main paper, we introduce the use of optimal transport (OT) to effectively handle these correlations. To illustrate this, we provide a visualization of the meaningful correlations captured by the OT approach. As depicted in Fig. 8, the visualization of the transport plan clearly shows that image and text pairs with direct semantic relationships are given priority. Conversely, pairs that are semantically irrelevant are typically disregarded by the transport plan.

# B  Additional Experiments

## B.1  Error bar analysis

Table 6: **Error bar analysis** on 18 image datasets.

| Method | OxfordFlowers | DTD | OxfordPets | StanfordCars | UCF101 | Caltech101 |
|---|---|---|---|---|---|---|
| CLIP | 67.44 | 44.27 | 88.25 | 65.48 | 65.13 | 93.35 |
| AWT | 74.42±0.09 | 55.30±0.03 | 92.30±0.34 | 69.83±0.10 | 72.45±0.26 | 94.94±0.17 |

| Method | Food101 | SUN397 | FGVCAircraft | EuroSAT | Birdsnap | Caltech256 | CUB |
|---|---|---|---|---|---|---|---|
| CLIP | 83.65 | 62.59 | 23.67 | 42.01 | 42.80 | 82.50 | 54.90 |
| AWT | 85.51±0.03 | 70.26±0.23 | 28.34±0.26 | 61.00±1.03 | 48.41±0.13 | 88.79±0.19 | 60.36±0.20 |

| Method | ImageNet-A | ImageNet | ImageNet-R | ImageNet-Sketch | ImageNet | OOD | Avg.(14) |
|---|---|---|---|---|---|---|---|
| CLIP | 47.74 | 60.75 | 73.98 | 46.13 | 66.74 | 57.15 | 63.06 |
| AWT | 60.42±0.22 | 64.86±0.21 | 80.27±0.09 | 51.64±0.03 | 71.33±0.02 | 64.30±0.03 | 69.52±0.08 |

We conducted an analysis of error bars across 18 image datasets, performing three runs each to ensure robust statistical evaluation. The results, which include both the mean and the standard deviation of the top-1 accuracy, are presented in Tab. 6. Overall, AWT demonstrates robust performance, exhibiting low variability across most datasets.

## B.2  Performance-efficiency trade-off

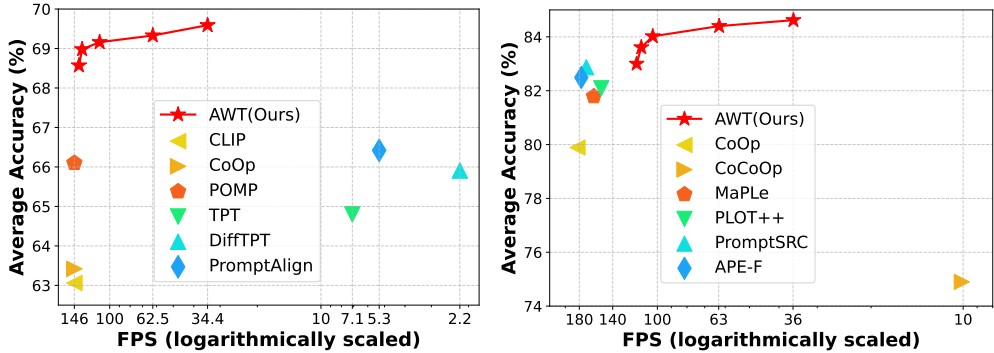

Figure 9: **Performance-efficiency comparison.** AWT is assessed using 2, 5, 10, 25, and 50 image views to present its trade-off between performance and computational efficiency. We show the results on both zero-shot (left) and few-shot (right) tasks.

Our AWT introduces additional computational costs during inference by utilizing multiple views. Although text embeddings can be pre-computed as suggested in [1], the primary expense arises from processing additional image views. In our study, we explore the trade-off between performance and efficiency by employing 2, 5, 10, 25, and 50 augmented views of the input image. We assess this trade-off by reporting both the frames per second (FPS) and the average accuracy in zero-shot and few-shot settings, as shown in Fig. 9.

The results indicate that even with a limited number of views, AWT can substantially outperform existing methods. As the number of views increases, there is a consistent enhancement in performance, albeit at the expense of reduced inference speed. This analysis enables practitioners to tailor AWT application to real-world scenarios by selecting an optimal balance between accuracy and processing speed, based on specific requirements.

In the few-shot settings, we incorporate two additional adapter modules at each transformer layer to facilitate transfer learning. This configuration initially makes our method slower than other prompt learning approaches, likely due to the adapters requiring sequential processing, whereas prompts are processed in parallel. Considering the computational costs, we propose that AWT could be more efficiently transferred using test-time zero-cost methods like LoRA [125], which allow for structural reparameterization after training.

## B.3 Study on the multi-modal adapter

Table 7: **Fine-tuned modalities in few-shot learning.** The model's performance is evaluated with the adapter module applied to visual, textual, or both modalities. Performance metrics are reported for the ImageNet test set.

| Adapter | 1shot | 4shot | 16shot |
|---|---|---|---|
| Visual | 71.66 | 72.52 | 74.34 |
| Textual | 71.24 | 72.99 | 74.54 |
| Multi-modal | **72.14** | **73.41** | **75.17** |

In this study, we explore the impact of applying multi-modal adapters to both visual and textual modalities. We conducted experiments using an unimodal adapter either on the visual or textual side and analyzed their performance on the ImageNet dataset. The results are detailed in Tab. 7. Our findings indicate that when training samples are extremely limited, such as in a 1-shot scenario, fine-tuning the visual adapter yields more benefits. However, as the number of training samples increases, fine-tuning the textual adapter becomes more advantageous. This shift is likely due to the need to reduce the similarity between class embeddings as suggested in [126]. Despite these variations, our results ultimately show that fine-tuning both modalities simultaneously delivers the best overall performance.

## B.4 Domain generalization

Table 8: **Domain generalization.** All methods are trained on 16-shot ImageNet and directly evaluated on four datasets with domain shifts. We report the top-1 accuracy (%) of each method.

| Method | IN-1K [89] | IN-A [103] | IN-V2 [104] | IN-R [105] | IN-K [106] | *OOD* |
|---|---|---|---|---|---|---|
| CoOp [9] | 71.51 | 49.71 | 64.20 | 75.21 | 47.99 | 59.28 |
| CoCoOp [13] | 71.02 | 50.63 | 64.07 | 76.18 | 48.75 | 59.91 |
| ProGrad [27] | 72.24 | 49.39 | 64.73 | 74.58 | 47.61 | 59.07 |
| KgCoOp [14] | 71.20 | 50.69 | 64.10 | 76.70 | 48.97 | 60.11 |
| PLOT++ [83] | 72.48 | 47.05 | 65.07 | 74.27 | 47.13 | 58.38 |
| MaPLe [15] | 70.72 | 50.90 | 64.07 | 76.98 | 49.15 | 60.28 |
| PromptSRC [15] | 71.27 | 50.90 | 64.35 | 77.80 | 49.55 | 60.65 |
| POMP [22] | 70.16 | 50.83 | 63.32 | 77.37 | 49.74 | 60.32 |
| TPT [49] + CoOp [9] | 73.61 | 57.95 | 66.83 | 77.27 | 49.29 | 62.83 |
| TPT [49] + CoCoOp [13] | 71.07 | 58.47 | 64.85 | 78.65 | 48.47 | 62.61 |
| DiffTPT [50] + CoOp [9] | 75.00 | 58.09 | 66.80 | 73.90 | 49.50 | 62.07 |
| DiffTPT [50] + CoCoOp [13] | 69.30 | 52.56 | 63.20 | 75.30 | 47.50 | 59.64 |
| PromptAlign [52] | 71.44 | 59.37 | 65.29 | 79.33 | 50.23 | 63.56 |
| AWT | **75.17** | **63.01** | **68.22** | **81.20** | **51.59** | **66.01** |

Previous studies [9, 13] suggested assessing method robustness by directly applying a model trained on ImageNet to four variant datasets derived from ImageNet. We adhere to this protocol and present our findings in Tab. 8. The results demonstrate that our approach, AWT, not only excels in few-shot performance on ImageNet but also outperforms in domain generalization robustness across the four ImageNet variant datasets, effectively balancing the adaptation-generalization trade-off.

## B.5 Study on different LLMs

Table 9: **Study on different LLMs in the description generation process.**

| AWT | IN-1K [89] | IN-A [103] | IN-V2 [104] | DTD [91] | Cars [93] | UCF [94] | Cal101 [95] | Food [96] |
|---|---|---|---|---|---|---|---|---|
| w/ GPT-3.5 [35] | 71.32 | **60.33** | 65.15 | 55.56 | 69.93 | **72.51** | **95.54** | **85.54** |
| w/ GPT-4o [36] | 71.36 | 60.13 | 64.63 | 56.03 | 69.46 | 72.14 | 95.21 | 85.39 |
| w/ Qwen-Plus [127] | **71.48** | 60.21 | **65.21** | **56.15** | **70.25** | 72.01 | 95.17 | 85.42 |

In this study, we utilize GPT-4o [36] and Qwen-Plus [127] for our description generation process, integrating them into AWT. The benchmark accuracy is presented in Tab. 9. Overall, AWT maintains robust performance across different LLMs. Interestingly, more advanced LLMs do not necessarily lead to significant performance improvements. This may be because AWT primarily relies on the simple instruction-following and knowledge-recall capabilities of LLMs, for which models like GPT-3.5 are sufficient.

## B.6 Description generation without dataset description

Table 10: **Compare AWT-`base` with other description generation methods.**

| Method | Manual Design Component | OOD | Avg.(14) |
|---|---|---|---|
| AWT w/ VisDesc [61] | 1 global question | 63.55 | 67.78 |
| AWT w/ CuPL-base [63] | 3 global questions | 63.57 | 68.87 |
| AWT w/ CuPL-full [63] | 4.56 questions per dataset | 63.88 | 69.33 |
| AWT-`base` | 1 global question | 64.03 | 69.27 |
| AWT | (1 global) + (1 description per dataset) | 64.43 | 69.59 |

Here, we examine the performance of AWT when no dataset-level description is available, a situation often encountered in real-world applications. We introduce a `base` version of AWT, which replaces our original instruction with `"Generate questions to classify images."` while leaving other components unchanged. The performance of AWT-`base` is detailed in Tab. 10. Despite its simplicity, AWT-`base` demonstrates favorable performance compared to previous methods.

## B.7 Numerical values in few-shot learning

For the benefit of future research, we present numerical values of our experimentation in the few-shot transfer learning setting in Tab. 11.

Table 11: **Few-shot image classification results** of different methods on 11 datasets. All results are averaged over three runs.

| | Shots | Pets[92] | Flowers[90] | Aircraft[98] | DTD[91] | SAT[99] | Cars[93] | Food[96] | SUN[97] | Cal101[95] | UCF[94] | IN-1k[89] | Avg.(11) |
|---|---|---|---|---|---|---|---|---|---|---|---|---|---|
| Handcrafted | - | 88.25 | 67.44 | 23.67 | 44.27 | 42.01 | 65.48 | 83.65 | 62.59 | 93.35 | 65.13 | 66.73 | 63.87 |
| CoOp [9] | 1 | 90.37 | 77.53 | 21.37 | 50.23 | 54.93 | 67.43 | 84.33 | 66.77 | 92.60 | 71.23 | 66.33 | 67.56 |
| | 2 | 89.80 | 87.33 | 26.20 | 53.60 | 65.17 | 70.50 | 84.40 | 66.53 | 93.07 | 73.43 | 67.07 | 70.65 |
| | 4 | 92.57 | 92.17 | 30.83 | 58.70 | 70.80 | 74.47 | 84.47 | 69.97 | 94.40 | 77.10 | 68.73 | 74.02 |
| | 8 | 91.27 | 94.97 | 39.00 | 64.77 | 78.07 | 79.30 | 82.67 | 71.53 | 94.37 | 80.20 | 70.63 | 76.98 |
| | 16 | 91.87 | 97.07 | 43.40 | 69.87 | 84.93 | 83.07 | 84.20 | 74.67 | 95.57 | 82.23 | 71.87 | 79.89 |
| CoCoOp [13] | 1 | 91.27 | 72.08 | 12.68 | 48.54 | 55.33 | 67.22 | 85.65 | 68.33 | 93.83 | 70.30 | 69.43 | 66.79 |
| | 2 | 92.64 | 75.79 | 15.06 | 52.17 | 46.74 | 68.37 | 86.22 | 69.03 | 94.82 | 73.51 | 69.78 | 67.65 |
| | 4 | 92.81 | 78.40 | 24.79 | 55.04 | 65.56 | 69.39 | 86.88 | 70.21 | 94.98 | 74.82 | 70.39 | 71.21 |
| | 8 | 93.45 | 84.30 | 26.61 | 58.89 | 68.21 | 70.44 | 86.97 | 70.84 | 95.04 | 77.14 | 70.63 | 72.96 |
| | 16 | 93.34 | 87.84 | 31.21 | 63.04 | 73.32 | 71.57 | 87.25 | 72.15 | 95.16 | 78.14 | 70.83 | 74.90 |
| PLOT++ [83] | 1 | 91.89 | 80.48 | 28.60 | 54.57 | 65.41 | 68.81 | 86.16 | 66.77 | 94.34 | 74.31 | 66.45 | 70.71 |
| | 2 | 92.29 | 89.81 | 31.14 | 56.72 | 76.80 | 73.17 | 86.33 | 68.06 | 94.69 | 76.76 | 68.28 | 74.00 |
| | 4 | 92.55 | 92.93 | 35.29 | 62.43 | 83.21 | 76.25 | 86.46 | 71.73 | 95.13 | 79.76 | 70.40 | 76.92 |
| | 8 | 93.02 | 95.44 | 41.42 | 66.49 | 88.37 | 81.26 | 86.58 | 73.93 | 95.51 | 82.80 | 71.31 | 79.65 |
| | 16 | 93.59 | 97.56 | 46.74 | 71.43 | 92.00 | 84.55 | 87.11 | 76.03 | 96.04 | 85.34 | 72.60 | 82.09 |
| PromptSRC [21] | 1 | 92.00 | 85.93 | 27.67 | 56.23 | 73.13 | 69.40 | 84.87 | 69.67 | 93.67 | 74.80 | 68.13 | 72.32 |
| | 2 | 92.50 | 91.17 | 31.70 | 59.97 | 79.37 | 73.40 | 85.70 | 71.60 | 94.53 | 78.50 | 69.77 | 75.29 |
| | 4 | 93.43 | 93.87 | 37.47 | 65.53 | 86.30 | 77.13 | 86.17 | 74.00 | 95.27 | 81.57 | 71.07 | 78.35 |
| | 8 | 93.50 | 96.27 | 43.27 | 69.87 | 88.80 | 80.97 | 86.90 | 75.73 | 95.67 | 84.30 | 72.33 | 80.69 |
| | 16 | 93.67 | 97.60 | 50.83 | 72.73 | 92.43 | 83.83 | 87.50 | 77.23 | 96.07 | 86.47 | 73.17 | 82.87 |
| ProVP-Ref [32] | 1 | 91.31 | 79.10 | 28.02 | 52.22 | 82.88 | 66.72 | 84.76 | 66.09 | 94.01 | 72.72 | 67.29 | 71.37 |
| | 2 | 92.31 | 86.11 | 30.55 | 57.23 | 86.57 | 67.93 | 85.54 | 68.42 | 95.09 | 76.07 | 68.54 | 74.03 |
| | 4 | 93.02 | 91.70 | 34.40 | 63.33 | 88.97 | 71.57 | 85.64 | 70.27 | 95.28 | 80.13 | 69.25 | 76.73 |
| | 8 | 93.27 | 95.89 | 43.47 | 68.42 | 91.78 | 77.33 | 86.41 | 72.48 | 95.86 | 83.57 | 70.23 | 79.88 |
| | 16 | 93.89 | 98.16 | 53.18 | 73.62 | 94.19 | 84.85 | 87.00 | 74.72 | 96.63 | 86.43 | 71.14 | 83.07 |
| APE-T [57] | 1 | 91.17 | 90.58 | 31.50 | 60.58 | 72.98 | 69.41 | 85.94 | 71.20 | 93.87 | 75.26 | 70.88 | 73.94 |
| | 2 | 91.77 | 94.15 | 35.85 | 62.94 | 76.73 | 73.77 | 86.01 | 72.19 | 94.93 | 79.17 | 71.43 | 76.27 |
| | 4 | 92.34 | 95.70 | 39.00 | 67.55 | 83.57 | 76.61 | 86.61 | 74.47 | 94.73 | 80.99 | 71.99 | 78.51 |
| | 8 | 93.46 | 96.91 | 42.87 | 70.63 | 87.02 | 78.86 | 86.85 | 76.13 | 95.70 | 83.72 | 73.37 | 80.50 |
| | 16 | 93.16 | 97.81 | 48.33 | 74.29 | 90.17 | 83.93 | 87.22 | 77.29 | 95.94 | 84.85 | 74.33 | 82.49 |
| **AWT** | 1 | 92.89 | 85.61 | 33.92 | 59.44 | 76.33 | 73.43 | 85.92 | 72.65 | 95.13 | 78.38 | 72.14 | 75.08 |
| | 2 | 92.98 | 89.89 | 37.77 | 62.63 | 83.17 | 76.96 | 86.05 | 73.72 | 95.67 | 80.32 | 72.86 | 77.46 |
| | 4 | 93.50 | 93.98 | 42.14 | 67.02 | 88.51 | 80.74 | 86.46 | 74.72 | 96.27 | 82.91 | 73.41 | 79.97 |
| | 8 | 93.83 | 96.47 | 50.21 | 70.51 | 89.72 | 84.50 | 87.44 | 75.79 | 96.58 | 85.42 | 74.38 | 82.26 |
| | 16 | 94.63 | 97.69 | 57.08 | 74.65 | 93.68 | 87.59 | 88.11 | 77.57 | 97.12 | 87.53 | 75.17 | 84.62 |

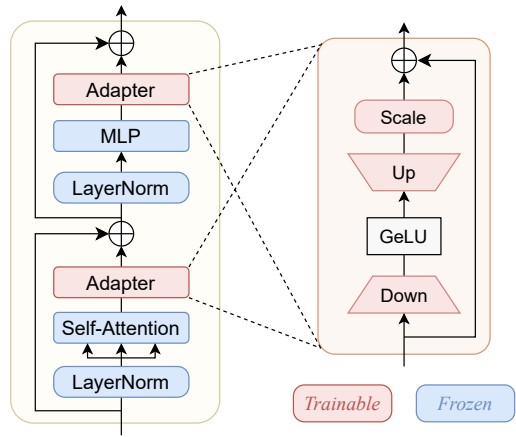

Figure 10: **Architecture of the adapter module** and its integration with the Transformer.

Table 12: **Few-shot transfer learning settings.** We select the model from the final epoch for testing.

| Configuration | 1-shot | 2-shot | 4-shot | 8-shot | 16-shot |
|---|---|---|---|---|---|
| adapter layers | | | all transformer layers | | |
| downsample rate $\lambda$ | 384 | 384 | 384 | 256 | 256 |
| learning rate | 2e-3 | 2e-3 | 2e-3 | 8e-3 | 8e-3 |
| learning rate schedule | | | cosine decay [128] | | |
| warmup epochs | | | 5 | | |
| warmup type | | | linear | | |
| weight decay | | | 5e-4 | | |
| training epochs | 20 | 20 | 20 | 25 | 25 |
| batch size | | | 16 | | |
| optimizer | | | SGD [129] | | |
| optimizer momentum | | | 0.9 | | |
| image views $N$ (train/test) | | | 8/50 | | |
| class descriptions $M$ (train/test) | | | 8/50 | | |
| GPU numbers | | | 1 | | |
| augmentation | | | random resized crop and random flip | | |

## C   Additional Details

### C.1   Few-shot learning with the multi-modal adapter module

For few-shot learning, we employ a multi-modal adapter to achieve parameter-efficient transfer learning. The architecture of the multi-modal adapter is depicted in Fig. 10. We integrate the adapter module subsequent to the Multi-Head Self-Attention and the MLP module within each transformer layer. The adapter module is applied to both the visual and textual side of the CLIP model. The core of the adapter module is a bottleneck structure designed to efficiently manage model parameters while maintaining performance. This structure comprises a down-projection layer $\boldsymbol{W}_{\text{down}} \in \mathbb{R}^{d \times \frac{d}{\lambda}}$ that reduces the dimensionality of the input features, and an up-projection layer $\boldsymbol{W}_{\text{up}} \in \mathbb{R}^{\frac{d}{\lambda} \times d}$ that restores the original dimensions. Between these layers, a GeLU activation layer [130] introduces non-linearity. A learnable scale parameter $s$ is used to modulate the output of the adapter. The feature dimension is denoted by $d$. For an input feature vector $\mathbf{x} \in \mathbb{R}^{L \times d}$, where $L$ is the sequence length, the adapter's forward process is expressed as:

$$\hat{\mathbf{x}} = s \cdot \text{GeLU}(\mathbf{x} \cdot \boldsymbol{W}_{\text{down}}) \cdot \boldsymbol{W}_{\text{up}} + \mathbf{x}. \tag{9}$$

Detailed settings and hyperparameters for our few-shot learning experiments are outlined in Tab. 12.

Table 13: **Dataset-level descriptions** employed in the first step of our prompt strategy are provided, along with corresponding source weblinks for each description. ImageNet-A and ImageNetV2 are omitted as they share the same dataset-level descriptions as ImageNet.

| Dataset | Description | Desc. Source |
|---|---|---|
| ImageNet | *"an image database contains millions of images across thousands of categories"* | [Website] |
| ImageNet-R | *"contains images in forms of: art, cartoons, deviantart, graffiti, embroidery, graphics, origami, paintings, patterns, plastic objects, plush objects, sculptures, sketches, tattoos, toys, and video game renditions"* | [Website] |
| ImageNet-Sketch | *"consists of black and white sketches of ImageNet categories"* | [Website] |
| Caltech101 | *"contains images from 101 object categories"* | [Website] |
| OxfordPets | *"a pet dataset whose images have a large variation in scale, pose, and lighting"* | [Website] |
| StanfordCars | *"contains images of cars whose classes are typically at the level of Make, Model, Year, ex"* | [Website] |
| OxfordFlowers | *"the flowers chosen to be flowers commonly occurring in the United Kingdom with large scale, pose, and light variations"* | [Website] |
| Food101 | *"consists of 101 food categories with some amount of noise"* | [Website] |
| FGVCAircraft | *"contains images of different aircraft model variants, most of which are airplanes"* | [Website] |
| SUN397 | *"a Scene UNderstanding dataset with 397 categories"* | [Website] |
| DTD | *"has collection of textural images in the wild"* | [Website] |
| EuroSAT | *"based on Sentinel-2 satellite images for land use and land cover classification"* | [Website] |
| UCF101 | *"an action recognition data set of realistic action videos"* | [Website] |
| Caltech256 | *"an object recognition dataset containing real-world images"* | [Website] |
| CUB | *"a challenging dataset of 200 bird species"* | [Website] |
| Birdsnap | *"a large bird dataset with 500 bird species"* | [Website] |

## C.2 Two-step dataset-aware prompting for LLMs

Here, we provide additional details of our proposed two-step dataset-aware prompting strategy for LLMs. Initially, in the first step, we leverage a dataset-level description for each dataset to inspire the generation of diverse and contextually relevant questions. The descriptions for each dataset are sourced from their official websites or from the paperswithcode website, and we include the dataset description and the reference to the source's URL in Tab. 13.

To better illustrate the effectiveness of our strategy, we present a sample of questions generated in the first step. For each dataset, we list three examples in Tab. 14. From these examples, we can observe the following: 1) the questions generated are not only more diverse compared to traditional prompts such as "Describe a {class}", but they also delve into more detailed inquiries, and 2) the questions incorporate dataset-specific information, thereby aiding LLMs in producing more accurate and relevant descriptions.

## C.3 License information of the assets used in this work

**Datasets.** Below are the datasets used in this paper that have known license information:
MIT License: ImageNet-A [103], ImageNetV2 [104], ImageNet-R [105], ImageNet-Sketch [106], EuroSAT [99].
CC BY 4.0 License: Caltech101 [95], Caltech256 [101], HMDB51 [107], K400 [119], K600 [108].
CC BY-SA 4.0 License: OxfordPets [92].

**Source code.** Source code used in this paper are under the MIT License: CLIP [1], CoOp [9], TPT [49], PLOT [83].

Table 14: **A selection of generated questions** for each dataset.

| Dataset | Generated questions (a selection) |
|---|---|
| ImageNet | How would you describe the overall appearance of a {} in the image? 
 Describe the color scheme and patterns present in the image of a {}. 
 What shapes and structures are noticeable in the image of a {}? |
| ImageNet-R | How is the style of the {} depicted in this art piece different from a realistic portrayal? 
 What aspects of the sculpture rendition of {} stand out compared to other forms? 
 What elements in the origami version of {} capture its essence in a creative way? |
| ImageNet-Sketch | What basic shapes can you identify in the sketch of {}? 
 How would you describe the linework in the sketch of {}? 
 Are there any particular lines or strokes that define the sketch of {}? |
| Caltech101 | How would you describe the overall appearance of {}? 
 Are there any recognizable patterns or markings on {}? 
 What colors dominate the image of {}? |
| OxfordPets | What fur patterns or textures are visible on {}? 
 Are there any distinctive features like ears, tails, or paws that help identify {}? 
 What colors are commonly seen on {} in the dataset? |
| StanfordCars | How would you describe the body shape of {}? 
 Are there any distinctive logos or emblems on {}? 
 What kind of wheels or tires are visible on {}? |
| OxfordFlowers | What are the prominent colors on the petals of {}? 
 Can you describe the shape of the petals on {}? 
 Do you notice any specific features like stamens or pistils on {}? |
| Food101 | What are the key ingredients or toppings present on {}? 
 Are there any garnishes or accompaniments that come with {}? 
 How would you describe the presentation or plating of {}? |
| FGVCAircraft | How would you describe the engine or propulsion system of the aircraft in {}? 
 What details can you observe in the cockpit area of the aircraft in {}? 
 Can you describe the wingspan of the aircraft in {}? |
| SUN397 | What kind of objects or structures are typically found in {}? 
 Are there any specific landmarks or features that are commonly associated with {}? 
 Can you identify any natural elements in {}? |
| DTD | How would you describe the surface of {}? 
 Are there any patterns or irregularities in the texture of {}? 
 Are there any specific features that indicate the material of {}? |
| EuroSAT | What natural features can you identify in a satellite view of {}? 
 What land patterns or formations are visible in a satellite view of {}? 
 Are there any specific landmarks that can help in identifying a satellite view of {}? |
| UCF101 | What body movements are being performed in the action of {}? 
 Can you describe the posture and gestures of the person in the action of {}? 
 What objects or tools are involved in the action of {}? |
| Caltech256 | Can you describe the overall shape of {}? 
 Are there any specific patterns or markings on {}? 
 How would you describe the color scheme of {}? |
| CUB | What specific colors can be seen on the feathers of {}? 
 Can you describe the beak shape of {}? 
 What are the distinguishing features of {}'s wings? |
| Birdsnap | How would you describe the shape of the body of {}? 
 What is the coloration of the feathers on {}? 
 Do you notice any distinct features on the head of {}? |

# D   Societal Impacts

This paper introduces AWT to enhance the transferability of vision-language foundation models. We have evaluated the effectiveness of AWT across a variety of classification tasks, employing different types, scales, and architectures of VLMs. Our findings suggest that AWT holds significant potential for integration into diverse downstream applications involving VLMs. Beyond classification tasks, we anticipate that the insights gained from this study will stimulate further research into other areas of computer vision, such as object detection and semantic segmentation. Currently, we are not aware of any ethical issues associated with the real-world applications of this technology. Nonetheless, continuous monitoring and evaluation are recommended to ensure responsible deployment.

# E   Limitations and Future Work

While the AWT framework achieves commendable performance using a limited number of augmented image views, attaining higher performance necessitates an increase in the number of views. This expansion, however, inevitably leads to additional computational costs. Notably, the visual elements in different augmented views often contain duplicates, particularly when they are all derived from the same original image (*e.g.*, random cropping). An interesting avenue for future research would be to explore strategies for reducing duplicates among batches to enhance inference speed. Furthermore, the application of test-time augmentation extends beyond our current scope and holds potential for various other domains, such as video understanding [131, 132], frame interpolation [133–135], semantic segmentation [136–138], and action detection [139]. Another intriguing area for future research involves the integration of efficient and controlled diffusion models [140, 141] to enhance the quality of visual augmentations.

