# OpenReview forum: "AWT: Transferring Vision-Language Models via Augmentation, Weighting, and Transportation"
_NeurIPS.cc/2024/Conference — NeurIPS 2024 poster_

### Official Review · Reviewer_qvMQ · 2024-07-07

**Soundness:** 3
**Presentation:** 3
**Contribution:** 3
**Rating:** 7
**Confidence:** 4

**Summary:**

This paper introduces an adaptation framework AWT to enhance vision-language models. The authors propose to augment inputs from visual and textual perspectives. The augmented inputs are dynamically reweighted based on the prediction entropy. Finally, the authors propose to use optimal transport to find the best prediction from VLM. The experimental results demonstrate strong performance under various settings.

**Strengths:**

- This paper is well illustrated and easy to follow.
- The proposed AWT framework is novel and effective, which improves performance of VLM without additional training costs.
- The experiments demonstrate the strong performance of AWT under image/video tasks.

**Weaknesses:**

- Performance-efficiency trade-off under few-shot settings. In supplementary material Figure 8, the performance increases with more augmented images under a zero-shot setting. I'd like to see a similar analysis under the few-shot image classification setting, which can further improve the significance of this work.
- How do the authors decide the hyper-parameters? The authors provide the average results of 14 datasets and the OOD under different choices of hyper-parameters in Table 4. However, it seems to lack details about how the hyper-parameters were chosen. Could authors provide more information on this?
- Why choose flipping as one of the image augmentation strategies? If I understand correctly, image augmentation aims to enable the VLM to capture the fine-grained matching between image regions and different text descriptions, which is already achieved by random crop.

**Questions:**

Please see the weakness above.

**Limitations:**

The limitation of this paper is sufficiently discussed by the authors.

---

> ### Author Rebuttal · Authors · 2024-08-02
>
> We sincerely appreciate your constructive feedback. Please allow us to address your concerns below.
>
> #### **W1: I'd like to see performance-efficiency trade-off under few-shot settings.**
>
> Thank you for your suggestion. We presented this experiment in Figure 2 of our attached PDF. In comparison to the zero-shot scenario, the few-shot settings incorporate two additional adapter modules at each transformer layer to facilitate transfer learning. This configuration results in our method being initially slower than other prompt learning approaches. This slowdown is likely due to the adapters needing sequential processing, whereas the prompts are processed in parallel. Taking into account the computational costs, we propose that AWT could be more efficiently transferred using test-time zero-cost methods like LoRA [1], which allows for structural reparameterization after training.
>
> #### **W2: How do the authors decide the hyper-parameters?**
>
> Thank you for your question. As detailed in Table 4 of our paper, AWT involves four main hyper-parameters: number of augmented image views ($N$), number of generated class descriptions ($M$), and the weighting temperatures $\gamma_v$ and $\gamma_t$.  We maintain consistent hyper-parameter values across all scenarios—whether zero-shot or few-shot, and for both image and video tasks.  This uniformity is due to our primary focus on the zero-shot capabilities, where typically no validation set is available for further tuning of hyper-parameters. Here is how we determined these values:
>
> - We used a subset of the ImageNet training set as the initial and only validation set for selecting hyper-parameters. We chose the hyper-parameter values based on the accuracy:
>   - For $\gamma_v$ and $\gamma_t$, we conducted a broad search and observed optimal performance around the value of 1/2, which we adopted as the default value.
>   - For $N$ and $M$, our selection process was similar, but we opted for the most cost-effective values rather than the absolute best. This decision was based on the observation that while increasing $N$ and $M$ can marginally improve results, it also significantly raises the computational costs associated with querying GPT.
>
> #### **W3: Why choose flipping as one of the image augmentation strategies?**
>
> You are correct in noting that cropping alone can capture fine-grained information. However, we believe that flipping introduces an additional layer of diversity in visual perspectives, which can potentially enhance the cross-modal matching process. To support our choice, we've conducted an ablation study detailed below. Our findings suggest a modest improvement, but given the minimal additional cost, we have decided to retain this method.
>
> | Augmentation                         |  OOD  | Avg.(14) |
> | ------------------------------------ | :---: | :------: |
> | Random resized cropping              | 64.22 |  69.45   |
> | Random resized cropping and flipping | 64.43 |  69.59   |
>
> ***We hope that these responses adequately address your queries and concerns. We look forward to your feedback and further guidance.***
>
> [1] Hu, Edward J., et al. "LoRA: Low-Rank Adaptation of Large Language Models." ICLR 2022.

---

> > ### Comment · Reviewer_qvMQ · 2024-08-09
> >
> > The authors' response addresses all my concerns. I will raise my score.

---

> > > ### Author Response · Authors · 2024-08-09
> > >
> > > Thank you for your positive recognition of our work!

---

### Official Review · Reviewer_kG9e · 2024-07-09

**Soundness:** 3
**Presentation:** 4
**Contribution:** 3
**Rating:** 7
**Confidence:** 5

**Summary:**

The paper considers a task of low-shot (zero-shot and few-shot) adaptation of vision-language models (VLMs). The proposed approach is based on diversifying the inputs of VLMs using augmentations for images and LLMs for class names. Each input is then weighted based on its prediction entropy, while optimal transport is employed to measure the distance between the test image and classes. The evaluation is performed across a wide range of datasets for different tasks, such as zero-shot classification and video action recognition, few-shot classification, and out-of-distribution generalization.

**Strengths:**

- One set of hyper-parameters is used across all datasets, which differs from most other methods that tune hyper-parameters per dataset, sometimes even per shot. This makes the proposed method really a low-shot one, as others tune hyper-parameters on a large annotated validation set.
- The paper tackles an important problem of adapting VLMs in low-shot settings. This is a very active area of research, and the proposed method is achieving state-of-the-art results across various benchmarks.
- The method is presented in a simple way that is easy to understand.

**Weaknesses:**

- Prompts for LLM to generate class descriptions contain a description of the dataset, which is a slight limitation. It would be beneficial to show the result for prompting without knowing the dataset description. CuPL and VisDesc comparisons in Table 4d are going in this direction; however, considering that the LLM version could be different, they are not a direct comparison.
- The paper shows that simple averaging of features from multiple image and text inputs does not work. However, a comparison with weighted averaging (weights based on prediction entropy) is missing. This would give a proper indication of how important optimal transport is in the method.
- The related work section could be significantly extended, e.g., methods based on adapters (CLIP-Adapter [1]) and based on memory (TIP-Adapter [2], APE [3]) are not mentioned at all. Additionally, a comparison with APE for few-shot classification would be beneficial.

*Minor comments:*

- Are $\gamma_v = 2$ and $\gamma_t = 2$ as in L217 or are the equal to $1/2$ as in tables 4e and 4f?

*References:*

[1] Peng Gao, Shijie Geng, Renrui Zhang, Teli Ma, Rongyao Fang, Yongfeng Zhang, Hongsheng Li, Yu Qiao, "CLIP-Adapter: Better Vision-Language Models with Feature Adapters", IJCV, 2024

[2] Renrui Zhang, Zhang Wei, Rongyao Fang, Peng Gao, Kunchang Li, Jifeng Dai, Yu Qiao, Hongsheng Li, "Tip-Adapter: Training-free Adaption of CLIP for Few-shot Classification", ECCV, 2022

[3] Xiangyang Zhu, Renrui Zhang, Bowei He, Aojun Zhou, Dong Wang, Bin Zhao, Peng Gao, "Not All Features Matter: Enhancing Few-shot CLIP with Adaptive Prior Refinement", ICCV, 2023

**After rebuttal**
The rebuttal has adequately addressed all of my concerns from the initial review.

**Questions:**

- What is the performance of the proposed method when using LLM prompts that do not include a description of the dataset?
- What is the performance of the method without optimal transport but using weighted averaging of features?

**Limitations:**

The authors have adequately addressed the limitations of the work.

---

> ### Author Rebuttal · Authors · 2024-08-02
>
> We sincerely appreciate your insightful feedback. Please allow us to address your concerns below.
>
> #### **W1: 1) The dataset description is a slight limitation. 2) Results for prompting without dataset description. 3) The LLM version for AWT, VisDesc [1], and CuPL [2] might differ.**
>
> Thank you for raising these issues.
>
> - **Dataset Description Limitation**: We acknowledge this limitation. However, to enable the LLM to perceive visual domains (e.g., sketches), manually inputting this prior information is necessary. CuPL (in its `full` version) addresses this by **manually designing multiple questions for each dataset**. We have **streamlined this process** by directly providing LLMs with dataset-level descriptions, enabling them to **automatically generate dataset-tailored questions**.
> - **Prompting without Dataset Description**: Your suggestion to explore scenarios without dataset descriptions is valuable, considering real-world applications where such information might not always be available. We have designed a "`base`" version for AWT where we replaced the instruction in Lines 146-147 with "Generate questions to classify images." while keeping other parts unchanged.
> - **LLM Version Consistency**: For VisDesc, we regenerated the descriptions using GPT-3.5 (the same version as used for AWT) using their official code. For CuPL, the descriptions were initially from SuS-X [3] and generated by GPT-3. We apologize for this oversight. To correct this, we have regenerated CuPL descriptions with GPT-3.5 and implemented both `base` and `full` versions of CuPL.
>
> In summary, we now provide additional results for AWT (`base`) and GPT-3.5-based CuPL (`base` and `full`) in the table below.
>
> | Method             | GPT version | Manual Design Component                  | OOD   | Avg.(14) |
> | ------------------ | ----------- | ---------------------------------------- | ----- | -------- |
> | AWT w/ VisDesc     | GPT-3.5     | 1 global question                        | 63.55 | 67.78    |
> | AWT w/ CuPL-`base` | GPT-3.5     | 3 global questions                       | 63.57 | 68.87    |
> | AWT w/ CuPL-`full` | GPT-3.5     | ~4.56 questions per dataset              | 63.88 | 69.33    |
> | AWT-`base`         | GPT-3.5     | 1 global question                        | 64.03 | 69.27    |
> | AWT                | GPT-3.5     | (1 global) + (1 descirption per dataset) | 64.43 | 69.59    |
>
> #### **W2: Results for weighted averaging are missing.**
>
> Thank you for pointing this out. We aimed to demonstrate the significance of OT by contrasting A(Img.+Name) and A+T in Table 4(a). We have also included the results for weighted averaging below for your reference.
>
> | Step  | OOD   | Avg.(14) |
> | ----- | ----- | -------- |
> | A+W   | 63.67 | 68.71    |
> | A+W+T | 64.43 | 69.59    |
>
> #### **W3: The related work section could be significantly extended.**
>
> We apologize for the oversight in our related work section. CLIP-Adapter [4] and Tip-Adapter [5] are pioneering studies proposing an adapter module and a training-free cache module for few-shot adaptation, respectively. APE [6] further leverages a CLIP prior for initialization in a more efficient manner. We will incorporate these studies into our related work section. Moreover, to enable a comparison between APE and AWT, we implemented APE-T (its highest-performing variant) using the CLIP-base/16 architecture across 11 few-shot learning datasets, utilizing the official code. We detail this comparison in Figure 1 of our attached PDF. APE-T will also be included in Figure 3 of our paper.
>
> #### **Minor: Are $\gamma_v=2$ and $\gamma_t=2$ or are the equal to 1/2 as in tables 4e and 4f?**
>
> We apologize for the confusion; both $\gamma_v$ and $\gamma_t$ are indeed set to 1/2.
>
> ***We hope that these responses adequately address your queries and concerns. We look forward to your feedback and further guidance.***
>
> [1] Menon, Sachit, and Carl Vondrick. "Visual Classification via Description from Large Language Models." ICLR 2023.
>
> [2] Pratt, Sarah, et al. "What does a platypus look like? generating customized prompts for zero-shot image classification." ICCV 2023.
>
> [3] Udandarao, Vishaal, Ankush Gupta, and Samuel Albanie. "Sus-x: Training-free name-only transfer of vision-language models." ICCV 2023.
>
> [4] Gao, Peng, et al. "Clip-adapter: Better vision-language models with feature adapters." IJCV 2024.
>
> [5] Zhang, Renrui, et al. "Tip-adapter: Training-free clip-adapter for better vision-language modeling." ECCV 2022.
>
> [6] Zhu, Xiangyang, et al. "Not all features matter: Enhancing few-shot clip with adaptive prior refinement." ICCV 2023.

---

> > ### Comment · Reviewer_kG9e · 2024-08-10
> > **Rebuttal reply**
> >
> > The rebuttal has adequately addressed all of my concerns from the initial review. As a result, I will raise my score to accept.

---

### Official Review · Reviewer_Vbjj · 2024-07-11

**Soundness:** 3
**Presentation:** 3
**Contribution:** 3
**Rating:** 6
**Confidence:** 4

**Summary:**

This paper proposes a novel adaptation framework for vision-language models, which replaces point-to-point alignment of text and images with set-to-set alignment. Specifically, the authors use image transformation and LLM generation to create augmentation sets. After weighting the augmentations based on entropy, optimal transport is used to compute the distance between vision and textual sets to measure their semantic correlation. Experiments are conducted on standard benchmarks.

**Strengths:**

1. Replacing point-to-point distance measurement with set-to-set distance measurement is a significant improvement in measuring semantic correlation.
2. The proposed method is well-motivated, with its three modules—augmentation, weight, and transport—being closely related, and their effects are ablated separately.
3. Experiments are conducted on a wide range of tasks.

**Weaknesses:**

1. PLOT [71] and Wang [73] also introduce optimal transport (OT) into vision-language models and are closely related to this work. The relationship and differences between these works need to be clearly clarified.
2. Some recent works should be acknowledged. For example, [a, b, c] have also been proposed for the adaptation of vision-language models. These works were accepted by CVPR 2024 and typically appeared on ArXiv in March, about two months before the NeurIPS submission deadline. Therefore, it is recommended to discuss these works as well.
3. The computational complexity should be analyzed and compared against prior methods.

[a] Dual Memory Networks: A Versatile Adaptation Approach for Vision-Language Models, CVPR2024

[b] MMA: Multi-Modal Adapter for Vision-Language Models, CVPR2024

[c] Efficient Test-Time Adaptation of Vision-Language Models, CVPR2024

**Questions:**

See weaknesses, especially the difference and connection with [71,73]

---

> ### Author Rebuttal · Authors · 2024-08-02
>
> We appreciate the time and effort you have dedicated to reviewing our paper. Please allow us to address your concerns below.
>
> #### **W1: Relationship and differences between prior OT methods and AWT.**
>
> Thanks for your comment. For a detailed response, please refer to our responses to common reviewer concerns. We will include this discussion in the final version of our paper.
>
> #### **W2: Some recent works should be acknowledged and discussed.**
>
> Thank you for bringing these excellent studies to our attention. We have carefully considered their relevance with AWT and plan to include the following discussions in our revised paper:
>
> - **DMN [1]** introduces a versatile adaptation framework suitable for various settings. However, DMN's applicability is constrained as it necessitates historical test data and is limited to scenarios where test and candidate classes are identical. AWT, while effective, also has its limitations: it requires additional LLMs and incurs higher computational costs. Noted that AWT and DMN could be potentially combined: DMN could utilize its capability to store AWT's augmented image features in memory alongside LLM augmented textual features, then replace the final dot product with our proposed entropy-based weighted OT distance.
> - **MMA [2]** introduces advanced adapter-based methods for VLMs. In zero-shot settings, MMA falls short compared to AWT due to its dependency on additional training data. In contrast, for few-shot adaptation/generalization settings, while AWT uses a standard adapter module, MMA introduces an advanced adapter that balances discrimination and generalization. Integrating MMA's adapter into AWT could potentially enhance performance.
> - **TDA [3]** proposes the use of a memory cache to mitigate the extensive computational demands of test-time adaptation methods. TDA is akin to the zero-shot adaptation version of DMN, which necessitates historical test data from identical candidate classes. Based on our analysis alongside DMN, our AWT method is also orthogonal to TDA, and integrating these two approaches could potentially enhance performance.
>
> #### **W3: The computational complexity should be analyzed and compared against prior methods.**
>
> Thank you for your suggestions. We have discussed the computational limitations in Sections B.2 and F of our Appendix, and compared our method with previous methods in Figure 8 of our paper. Below, we provide additional details to more comprehensively address your concern:
>
> - **Augment and CLIP forward:** Given that the text features can be pre-computed (as suggested in the CLIP paper), AWT theoretically involves computing complexity of $N+1$ times that of the original CLIP model, where $N$ is the number of augmented image views.
> - **Weight Calculation:** This process is relatively swift and primarily involves several matrix operations such as Softmax function.
> - **Optimal Transport:** Thanks to the efficient Sinkhorn’s Algorithm, we can solve the optimal transport problem with high efficiency.
>
> The bulk of the computational effort is concentrated in the CLIP forward process. Thanks to the parallel processing capabilities of modern GPUs and efficient deep learning libraries like PyTorch, the computational cost of AWT remains manageable. Notably, AWT is significantly more efficient than test-time adaptation methods, which require model training during inference, such as TPT [4] (refer to Figure 8 of our paper). Moreover, the cost-performance trade-off can be easily modulated by adjusting $N$. We believe that these limitations can be further addressed through future research, as discussed in Lines 776-779 of our paper.
>
> ***We hope that these responses adequately address your queries and concerns. We look forward to your feedback and further guidance.***
>
> [1] Zhang, Yabin, et al. "Dual memory networks: A versatile adaptation approach for vision-language models." CVPR 2024.
>
> [2] Yang, Lingxiao, et al. "MMA: Multi-Modal Adapter for Vision-Language Models." CVPR 2024.
>
> [3] Karmanov, Adilbek, et al. "Efficient Test-Time Adaptation of Vision-Language Models." CVPR 2024.
>
> [4] Shu, Manli, et al. "Test-time prompt tuning for zero-shot generalization in vision-language models." NeurIPS 2022.

---

> > ### Comment · Reviewer_Vbjj · 2024-08-12
> >
> > Thanks for your responses. My concerns have been addressed and I will increase my score to 6.

---

### Official Review · Reviewer_CYNL · 2024-07-12

**Soundness:** 3
**Presentation:** 3
**Contribution:** 2
**Rating:** 4
**Confidence:** 3

**Summary:**

This paper proposes AWT, an adaptation framework that can boost pre-trained vision-language models (VLMs) for understanding new concepts on new classes. To summarize, AWT first do augmentations on both visual images and textual class names. Then an entropy metric is employed for weighting multiple augmented data. Finally, it measures the distance by OT algorithm associated with the confidence weight computed from the W step. AWT performances best in multiple scenarios including ZS, FS image classification and action recognition and OOD generalization.

**Strengths:**

1.This paper is well-written, and well-organized with clear statements, standard structure, intuitive motivation, etc.
2.AWT shows superior performance and strong generalization ability across multiple evaluation benchmarks.
3.Utilizing LLM to augment existing class names seems interesting and effective, also distinctive from previous works. May bring new insight for the community.

**Weaknesses:**

1.Augmenting images by simple random resized cropping and random flipping seems to be too naïve compared with nowadays prompting methods. As in previous work SuS-X [1], a much more diversified Support Set can be constructed by stable diffusion and retrieval. Meanwhile, according to Tab 4(a), we can not see any improvement by introducing visual augmentations (Raw inputs vs. A(Img.)).
[1] SuS-X: Training-Free Name-Only Transfer of Vision-Language Models. Vishaal Udandarao et al.
2.The final inference step, measuring distance across two modalities by OT algorithm is not new as it was proposed in [2] and [3]. However, the difference with these two studies is not clearly claimed in the AWT paper. The original statement “Distinct from these two studies, our research diverges by eschewing the need for additional training resources, opting instead for an augmentation-based direction”  looks confused and fuzzy.
[2] Tuning Multi-mode Token-level Prompt Alignment across Modalities. Dongsheng Wang et al.
[3] PLOT: Prompt Learning with Optimal Transport for Vision-Language Models. Guangyi Chen et al.

**Questions:**

1.What about applying more complex visual augmentations more than random cropping and flipping? How about introducing additional visual training data like Support Set in SuS-X?
2.Why only 50 class descriptions M are generated? The performance seems not saturated at M=50 according to Tab 4(c).
3.Any inference speed limitation? Since 50 times more images and class descriptions are generated by augmentation.
4.Are the evaluation results stable or constant? If AWT generate different image crops or different class descriptions (affected by GPT version) in multiple experimental trials, then the weighting and OT processes should be affected, which may give different final distance measure. If so, then the mean and variance of several rounds of experiments should be reported.

**Limitations:**

No limitation discussion in the original paper.

---

> ### Author Rebuttal · Authors · 2024-08-02
>
> Thank you for your detailed review and insightful comments! Please allow us to address your concerns below.
>
> #### **W1 and Q1: 1) Random resized cropping and flipping is naïve. 2) Cannot see any improvement by visual augmentations. 3) What about applying more complex visual augmentations?**
>
> Thank you for raising these points.
>
> - We opted for random resized cropping and flipping primarily due to their **efficiency** when compared to more advanced augmentation methods. For instance, employing diffusion models can significantly slow down inference speed, as discussed in the last part of Section 4.3 in DiffTPT [1], where inferring 10 test images takes 36 minutes. Meanwhile, methods involving data retrieval often require substantial storage space, such as LAION-5B used in SuS-X [2].
>
> - The incremental improvements with visual augmentations are also noted by us, as mentioned in Lines 277-278 of our paper. Directly applying such augmentations can generate information-sparse image views. **This motivates us to propose the weighting strategy**. We provided an additional ablation study below, which demonstrates that, with the weighting module, even simple visual augmentations can yield significant improvements.
>
> | Step             |  OOD  | Avg.(14) |
> | ---------------- | :---: | :------: |
> | Raw inputs       | 57.15 |  63.06   |
> | A(Img.)          | 56.77 |  63.40   |
> | A(Img.) + Weight | 62.76 |  66.21   |
>
> - We have explored more complex augmentations like diffusion models in Appendix Section D and found them superior to naive methods. Additionally, we included results for the EuroSAT dataset this time, summarized in the table below. However, given the efficiency concerns, we choose to use simple augmentation by default.
>
> | Augmentation                           | EuroSAT | CIFAR10 | CIFAR100 |
> | -------------------------------------- | :-----: | :-----: | :------: |
> | Random resized crop and flip (default) |  58.61  |  89.25  |  67.61   |
> | DALL·E 2                               |  60.41  |  92.30  |  68.69   |
>
> #### **W2: Difference between prior OT methods and AWT is not clearly claimed.**
>
> We apologize for any confusion caused. For a detailed response, please refer to our responses to common reviewer concerns. We will ensure to clarify any ambiguities in the final version of our paper.
>
> #### **Q2: Why only 50 class descriptions are generated?**
>
> Thank you for your query. As detailed in Table 4(c), increasing the number of descriptions from 50 to 100 provides only marginal gains (0.03% and 0.14% improvements, respectively), while doubling the query costs for GPT. We believe that maintaining 50 descriptions strikes a good cost-effectiveness balance.
>
> #### **Q3: Any inference speed limitation?**
>
> Thank you for your question. We have discussed the inference speed limitations in Sections B.2 and F of our Appendix.
>
> #### **Q4: Are the evaluation results stable or constant?**
>
> The results are stable and almost constant. The stability of AWT is supported by error bar analysis (mean and variance) in Section B.1 of the Appendix. For each run, images are augmented using different seeds, and descriptions are regenerated. Your suggestion to explore the impact of different GPT versions is insightful. Following your advice, we conducted further experiments with an alternative GPT version and another LLM, and presented the results below. Overall, AWT maintains robust performance across these variations. An interesting observation is that more advanced LLMs do not necessarily yield significant performance improvements. This may be attributed to the fact that, in this scenario, AWT primarily relies on the simple instruction-following and knowledge-recall capabilities of LLMs, making models like GPT-3.5 sufficient.
>
> | Method                   | ImageNet | ImageNet-A | ImageNet-V2 |  DTD  | StanfordCars | UCF101 | Caltech101 | Food101 |
> | ------------------------ | :------: | :--------: | :---------: | :---: | :----------: | :----: | :--------: | :-----: |
> | AWT w/ GPT-3.5 (default) |  71.32   |   60.33    |    65.15    | 55.56 |    69.93     | 72.51  |   95.54    |  85.54  |
> | AWT w/ GPT-4o            |  71.36   |   60.13    |    64.63    | 56.03 |    69.46     | 72.14  |   95.21    |  85.39  |
> | AWT w/ Qwen-Plus         |  71.48   |   60.21    |    65.21    | 56.15 |    70.25     | 72.01  |   95.17    |  85.42  |
>
> #### **L1: No limitation discussion in the original paper.**
>
> Thank you for your comment. The limitations have been discussed in Section F of the Appendix.
>
> ***We hope that these responses adequately address your queries and concerns. We look forward to your feedback and further guidance.***
>
> [1] Feng, Chun-Mei, et al. "Diverse data augmentation with diffusions for effective test-time prompt tuning." ICCV 2023.
>
> [2] Udandarao, Vishaal, Ankush Gupta, and Samuel Albanie. "Sus-x: Training-free name-only transfer of vision-language models." ICCV 2023.

---

> ### Author Response · Authors · 2024-08-12
>
> Dear Reviewer CYNL,
>
> We greatly appreciate the time and effort you have dedicated to reviewing our submission. As the discussion phase is set to conclude in 48 hours, we wish to ensure that all your queries and concerns are thoroughly addressed. Please do not hesitate to reach out if there are additional aspects of our submission that you would like us to clarify or elaborate on.
>
> Thank you once again for your insightful feedback.
>
> Best regards,
>
> Authors of Paper #8459

---

> ### Author Response · Authors · 2024-08-14
>
> Dear Reviewer CYNL,
>
> We would like to remind you that there are only 4 hours remaining in the discussion phase. Please let us know if all your concerns and questions have been addressed satisfactorily.
>
> We look forward to your feedback.
>
> Best regards,
>
> Authors of Paper #8459

---

### Author Rebuttal · Authors · 2024-08-02

Dear Reviewers,

We sincerely appreciate the time and effort you have invested in reviewing our submission. This paper received 4 review comments, and 3 reviewers gave positive scores. All reviewers acknowledged the strong writing, the excellent benchmark performance, and the extensive experiments conducted. Additionally, Reviewer CYNL noted that our study may bring new insights to the community, Reviewer Vbjj highlighted our significant improvements in measuring semantic correlation, Reviewer kG9e recognized that our paper addresses an important issue in low-shot learning, and Reviewer qvMQ commended the novelty and clarity of our method. We sincerely thank all reviewers for their recognition and valuable insights!

Below, we first address a common concern raised by the reviewers, followed by detailed responses to each reviewer's specific comments. We hope that our clarifications help enhance the understanding of our work.

#### **Common concern: Relationship and differences between prior works (PLOT [1], Wang et al. [2]) and AWT.**

**Relationship:** Both [1] and [2] lay the groundwork for AWT by demonstrating that the cross-modality distance can be effectively measured using Optimal Transport (OT).

**Differences:** AWT distinguishes itself from [1] and [2] in several key aspects:

- **Purpose:** While [1] and [2] utilize OT to learn comprehensive and semantically diverse prompts **during training**, AWT leverages OT to **directly enhance VLMs during inference**.
- **Elements for OT:** Both [1] and [2] apply OT at the **token level**, requiring **fine-grained alignment capabilities** that are often lacking in models like CLIP. This necessitates additional training data to close the gap. In contrast, AWT implements OT at the **holistic level**, treating entire images or sentences as single elements, thus eliminating the need for extra data.
- **Element Distributions:** Both [1] and [2] consider elements as **uniform distributions**, which may not be appropriate. AWT, on the other hand, uses real-time prediction entropy to **dynamically adjust distribution probabilities**. Our ablation studies support that relying on uniform distributions often leads to sub-optimal results (refer to Tables 4e and 4f, Lines 299-304 in our paper).

Consequently, **AWT significantly outperforms both methods** in zero-shot image classification tasks **without any training**. Below, we present a comparative analysis:

| Method          | Train |  Flowers  |    DTD    | OxfordPets | StanfordCars |  UCF101   | Caltech101 |  Food101  |  SUN397   | FGVCAircract |  EuroSAT  |
| --------------- | :---: | :-------: | :-------: | :--------: | :----------: | :-------: | :--------: | :-------: | :-------: | :----------: | :-------: |
| PLOT++  [1]     |   √   |   69.10   |   38.42   |   90.49    |    61.20     |   68.94   |   91.32    |   86.07   |   61.59   |    24.84     |   49.90   |
| Wang et al. [2] |   √   |   73.75   |   46.75   |   90.55    |    65.84     |   69.60   |   93.91    | **86.40** |   67.59   |    24.95     |   47.25   |
| AWT             |   ×   | **75.07** | **55.56** | **92.53**  |  **69.93**   | **72.51** | **95.54**  |   85.54   | **70.58** |  **29.22**   | **58.61** |

[1] Chen, Guangyi, et al. "PLOT: Prompt Learning with Optimal Transport for Vision-Language Models." ICLR 2023.

[2] Wang et al., Dongsheng, et al. "Tuning multi-mode token-level prompt alignment across modalities." NeurIPS 2023.

---

### Decision · Program_Chairs · 2024-09-25

**Decision:**

Accept (poster)

**Comment:**

This work involves the combination of three distinct, already understood, steps to improve the performance of vision-language models on classification centric tasks. The reviewers were broadly supportive of the paper, and agreed that while the methodological novelty was not very high the method performs very well when compared to existing baselines. The evaluation is thorough and the description of the method and analysis of the results is clear.

Given the above, this AC agrees with the reviewers and recommends that the paper is accepted. The authors are strongly encouraged to take the reviewer recommendations and suggestions into account when revising the paper for the camera ready version. Specifically, they should focus on the following:
* Add the comparisons flagged by the reviewers, e.g. PLOT, Wang et al, CLIP-Adapter, TIP-Adapter, etc.
* Make reference to Section C2 in the main paper so that it is clearer to the reader why dataset specific information is used by the LLM. Also add the new results from the rebuttal related to dataset descriptions to the appendix.
* Make the additional inference cost clearer in the results text e.g. point to Fig 8 in this section and add more of the methods from the main results table to the plot. It should be made very clear in the main text how many forward passes you need, and how that is more than some existing methods.
* Add the results from Fig 2 from the rebuttal PDF to the appendix.